# Sparsity-Driven Plasticity in Multi-Task Reinforcement Learning

**Aleksandar Todorov**[†][*]                               *a.todorov.4@student.rug.nl*

**Juan Cardenas-Cartagena**[†]                          *j.d.cardenas.cartagena@rug.nl*

**Rafael F. Cunha**[†]                                    *r.f.cunha@rug.nl*

**Marco Zullich**[†]                                      *m.zullich@rug.nl*

**Matthia Sabatelli**[†]                                  *m.sabatelli@rug.nl*

[†]*University of Groningen, Groningen, The Netherlands*

**Reviewed on OpenReview:** *https://openreview.net/forum?id=9L4Z23EfE9*

## Abstract

Plasticity loss, a diminishing capacity to adapt as training progresses, is a critical challenge in deep reinforcement learning. We examine this issue in multi-task reinforcement learning (MTRL), where higher representational flexibility is crucial for managing diverse and potentially conflicting task demands. We systematically explore how sparsification methods, particularly Gradual Magnitude Pruning (GMP) and Sparse Evolutionary Training (SET), enhance plasticity and consequently improve performance in MTRL agents. We evaluate these approaches across distinct MTRL architectures (shared backbone, Mixture of Experts, Mixture of Orthogonal Experts) on standardized MTRL benchmarks, comparing against dense baselines, and a comprehensive range of alternative plasticity-inducing or regularization methods. Our results demonstrate that both GMP and SET effectively mitigate key indicators of plasticity degradation, such as neuron dormancy and representational collapse. These plasticity improvements often correlate with enhanced multi-task performance, with sparse agents frequently outperforming dense counterparts and achieving competitive results against explicit plasticity interventions. Our findings offer insights into the interplay between plasticity, network sparsity, and MTRL designs, highlighting dynamic sparsification as a robust but context-sensitive tool for developing more adaptable MTRL systems.

## 1 Introduction

Although deep reinforcement learning (DRL) agents have demonstrated impressive results in various applications (Levine et al., 2016; Silver et al., 2017; Bellemare et al., 2020; Mathieu et al., 2023), these achievements come with notable trade-offs. Attaining state-of-the-art performance often relies on large-scale computational resources and heavily overparameterized models (Botvinick et al., 2019; Glanois et al., 2022; Thompson et al., 2022), which may lead to agents that either generalize poorly (Kirk et al., 2023) or struggle to adapt to new tasks or data over time. The former issue is a topic of interest within the transfer learning literature (Farebrother et al., 2020; Sabatelli & Geurts, 2021; Sasso et al., 2023; Zhu et al., 2023), while the latter is commonly referred to as plasticity loss (Nikishin et al., 2022; Lyle et al., 2023; Dohare et al., 2024). Plasticity loss manifests through several interconnected optimization pathologies: gradient interference leading to premature convergence (Lyle et al., 2024a), representational collapse, limiting the diversity of learned features

---

*Corresponding author

(Moalla et al., 2024), and neuronal saturation or dormancy that reduces effective network capacity (Bjorck et al., 2021; Sokar et al., 2023). While these challenges have been primarily investigated within single-task RL, (Nikishin et al., 2023; Abbas et al., 2023; Klein et al., 2024; Nauman et al., 2024a; Dohare et al., 2024), in this paper, we study them under the lens of multi-task reinforcement learning (MTRL), where maintaining representational flexibility across diverse tasks with potentially conflicting demands is even more crucial (Teh et al., 2017; Sodhani et al., 2021; D'Eramo et al., 2024). It naturally follows that this increased need for dynamic adaptation can make MTRL agents especially vulnerable to plasticity loss, as networks must simultaneously accommodate varied objectives without experiencing negative task interference (Liu et al., 2023). The necessity of determining which knowledge to share across tasks, and how to share it without harmful interference, further complicates the learning process (Devin et al., 2016; Sasso et al., 2023). Moreover, this challenge can be even further exacerbated by inefficient use of network capacity (Kumar et al., 2021), with significant portions of large networks becoming underutilized during training, ultimately hindering the acquisition of a universal policy capable of addressing multiple tasks concurrently. Recent work in neural network pruning offers a promising direction beyond mere compression, showing that sparse agents can match or even exceed dense counterparts in single-task RL (Livne & Cohen, 2020; Graesser et al., 2022; Obando-Ceron et al., 2024). Notably, methods like Gradual Magnitude Pruning (GMP) have shown positive effects on both single-task performance and plasticity (Obando-Ceron et al., 2024). Similarly, dynamic sparse training methods such as Sparse Evolutionary Training (SET) (Mocanu et al., 2018) have also proven effective in single-task RL, offering an alternative way to maintain and adapt sparsity throughout training (Graesser et al., 2022). These successes suggest that such sparsification approaches could address the optimization pathologies that undermine effective multi-task learning. Nonetheless, a systematic investigation of their impact on MTRL, where representational flexibility demands are significantly higher (Devin et al., 2016), remains largely unexplored.

This paper investigates whether sparsification methods, specifically GMP and SET (Mocanu et al., 2018), can enhance plasticity in MTRL agents, thereby improving performance across multiple tasks simultaneously. Our choice to explore these methods is motivated by their demonstrated success in single-task settings and the need to understand their efficacy in the MTRL domain. We evaluate this across various multi-task architectures, including shared backbones with task-specific heads (MTPPO), Mixture of Experts (MoE) (Ceron et al., 2024), and Mixture of Orthogonal Experts (MOORE) (Hendawy et al., 2024), using common MTRL benchmarks that range from partially observable environments with sparse reward to high-dimensional and continuous state and action spaces. Our central aim is to understand if the benefits of pruning can be primarily attributed to the mitigation of key plasticity loss indicators. Our experiments compare sparse agents against dense baselines, less adaptive sparsification techniques, and a suite of alternative plasticity-inducing or regularization methods, including Layer Normalization (Ba et al., 2016; Lyle et al., 2024a), ReDo (Sokar et al., 2023), Reset (Ash & Adams, 2020; Nikishin et al., 2022), and Weight Decay.

Our main contributions are, therefore, threefold:

- We establish that sparsification methods, particularly Gradual Magnitude Pruning (GMP) and Sparse Evolutionary Training (SET), serve as effective mechanisms for mitigating key indicators of plasticity degradation in MTRL, such as neuron dormancy and representational collapse. While the extent of these benefits varies with network architecture, sparse agents, especially in MTPPO and MoE configurations, consistently exhibit improved plasticity profiles compared to their dense counterparts.

- We empirically demonstrate that these plasticity improvements induced by sparsification often correlate with enhanced multi-task performance. Sparse agents frequently outperform dense baselines and demonstrate competitive performance against alternative, specialized methods explicitly designed to induce plasticity, as well as common regularization techniques.

- We show that the impact of sparsification on both plasticity and performance is architecture-dependent, offering insights into the interplay between network design, sparsity, and learning dynamics. This highlights sparsification as a valuable but context-sensitive tool in the MTRL toolkit. Furthermore, we highlight that beyond performance, sparsification offers inherent advantages such as potential for computational efficiency and a distinct form of implicit regularization not fully replicated by common regularization methods.

## 2 Background

This section provides the necessary context for our approach. We begin by outlining the mathematical preliminaries underlying our framework, including key concepts and notations from reinforcement learning. Subsequently, we review related work, focusing on recent advances in sparsity in deep reinforcement learning, plasticity loss, and multi-task learning.

### 2.1 Preliminaries

We consider the Partially Observable Markov Decision Process (POMDP), defined by a tuple $(\mathcal{S}, \mathcal{A}, \mathcal{P}, \mathcal{R}, \Omega, \mathcal{O}, \gamma)$, consisting of a state space $\mathcal{S}$, an action space $\mathcal{A}$, transition dynamics $\mathcal{P} : \mathcal{S} \times \mathcal{A} \to \Delta(\mathcal{S})$, a reward function $\mathcal{R} : \mathcal{S} \times \mathcal{A} \to \mathbb{R}$, observation space $\Omega$, observation probability function $\mathcal{O} : \mathcal{S} \times \mathcal{A} \to \Delta(\Omega)$, and discount factor $\gamma \in [0, 1)$. At each timestep $t$, the agent is situated in the true state $s_t \in \mathcal{S}$ and performs an action $a_t \in \mathcal{A}$. This causes the agent to transition to a new state $s_{t+1} \in \mathcal{S}$, receiving an observation $o_{t+1} \in \Omega$, and a reward $r_{t+1} = \mathcal{R}(s_t, a_t)$. The objective is to learn a policy $\pi_\theta(a_t|o_t)$ with parameters $\theta$ that maximizes the expected sum of discounted future rewards $J(\theta)$. In MTRL, the agent must learn a policy for a distribution of tasks $\mathcal{T}$. We adopt the Block Contextual POMDP framework (Sodhani et al., 2021; Hendawy et al., 2024), defined as $(\mathcal{C}, \mathcal{S}, \mathcal{A}, \mathcal{M}')$, where $\mathcal{C}$ represents the contextual space such that $c \in \mathcal{C}$ identifies a specific task $\tau \sim \mathcal{T}$. The mapping $\mathcal{M}'(c)$ provides the task-specific POMDP components $\{\mathcal{R}^c, \mathcal{P}^c, \mathcal{S}^c, \Omega^c, \mathcal{O}^c, \gamma^c\}$. The policy is now conditioned on the current observation $o \in \Omega^c$ and task context $c \in \mathcal{C}$. The objective is to maximize the expected return across all tasks $\mathbb{E}_{\tau \sim \mathcal{T}} [J_\tau(\theta)]$.

### 2.2 Related Work

**Sparsity in Reinforcement Learning** While deep reinforcement learning has traditionally relied on overparameterized networks, recent research challenges the necessity of such scale, suggesting that sparse networks can match or even exceed the performance of dense models (Livne & Cohen, 2020; Graesser et al., 2022). This trend highlights that DRL agents often underutilize their capacity (Kumar et al., 2021) or overfit to early experiences (Nikishin et al., 2022), making them particularly amenable to the regularizing effects of sparsity. Pruning neural connections reduces model complexity and noise, offering a form of structural regularization that can improve robustness and generalization (Jin et al., 2022). Our work investigates how sparsity-based methods can influence learning dynamics and plasticity in multi-task RL.

**Plasticity Loss in Reinforcement Learning** It is well known that Reinforcement learning systems face a unique form of non-stationarity stemming from evolving policies, shifting data distributions, and the bootstrapping nature of value updates. This can culminate in the form of plasticity loss, a reduced ability of the network to learn from new experiences, even within familiar data distributions (Lyle et al., 2022; Dohare et al., 2024). Plasticity loss often manifests as premature performance plateaus, training instability, and heightened sensitivity to hyperparameter settings (Igl et al., 2021; Berariu et al., 2023; Klein et al., 2024). Current understanding attributes plasticity loss primarily to unstable learning targets that create challenging optimization landscapes, with associated symptoms like collinear gradients (Lyle et al., 2024a), representational collapse (Moalla et al., 2024), and volatile gradient norms under adaptive optimizers (Lyle et al., 2024b). Internally, networks may suffer from shifting activation distributions, neuron saturation, or increasing dormancy over time (Sokar et al., 2023; Bjorck et al., 2021). Several interventions have been proposed to mitigate these effects. These include resetting techniques, such as periodic last-layer reinitialization (Ash & Adams, 2020; Nikishin et al., 2022; 2023), parameter update modulation strategies like Hare and Tortoise networks (Lee et al., 2024), and various architectural or optimization-based approaches, including weight decay (Sokar et al., 2023), deep Fourier features (Lewandowski et al., 2024a), and classification-based value learning (Farebrother et al., 2024). Normalization layers have also demonstrated benefits in this regard (Bhatt et al., 2023). Notably, Obando-Ceron et al. (2024) showed that gradual pruning can outperform many methods specifically designed to promote plasticity. This supports the broader notion that general-purpose regularization might offer a more robust and simpler solution to plasticity loss than domain-specific mechanisms, reinforcing the broader lesson that simplicity often outperforms specialized interventions (Klein et al., 2024; Nauman et al., 2024a).

**Multi-Task Reinforcement Learning** MTRL seeks to train a single agent across multiple tasks, balancing knowledge sharing for transfer against the risk of negative interference. Common MTRL techniques include shared encoders with task-specific heads (Teh et al., 2017), modular network designs (Yang et al., 2020), reward normalization (Hessel et al., 2018), compositional policy learning (Sun et al., 2022), and gradient projection or masking strategies (Yu et al., 2020; Hendawy et al., 2024). Mixture-of-Experts models have also gained traction, often enhanced with attention or orthogonality constraints for better task separation (Ceron et al., 2024; Cheng et al., 2023). Similarly, maintaining weight matrix orthogonality through regularization has been explored to enhance plasticity in continual learning settings, which face similar challenges to MTRL (Chung et al., 2024). A key observation in MTRL is that, unlike trends in supervised learning, simply scaling model capacity does not inherently guarantee performance improvements (Hansen et al., 2023; Ceron et al., 2024; Nauman et al., 2024b). Supervised approaches like SimBa, for instance, suggest that gains from scaling require careful inductive biases (Lee et al., 2025a). In contrast, network sparsity has demonstrated improvements in both generalization and plasticity in RL without necessarily relying on increased model scale (Graesser et al., 2022; Obando-Ceron et al., 2024). Despite the promise of sparsity, its implications in MTRL have remained largely unexplored. To the best of our knowledge, this work is the first to systematically examine how different sparsity-inducing techniques affect performance and plasticity in the multi-task regime. We aim to fill this gap by investigating how pruning and sparse connectivity can mitigate plasticity loss and promote stable, generalizable learning in complex task environments.

## 3 Experimental Setup

Our experiments systematically compare the effects of different sparsification approaches against dense baselines and two families of plasticity-enhancing interventions. The first family comprises explicit plasticity-restoring techniques that directly intervene on the agent's parameters to counteract plasticity loss. These include ReDo (Sokar et al., 2023), which periodically resets dormant neurons based on activity thresholds, and Reset (Ash & Adams, 2020; Nikishin et al., 2022), which reinitializes specific network layers at fixed intervals to combat primacy bias. The second family involves more implicit regularization-based mechanisms, which do not directly manipulate network dynamics but are known to stabilize training and encourage generalization. Specifically, we evaluate standard Weight Decay (WD) applied to dense agents, and Layer Normalization (LayerNorm) (Ba et al., 2016), which has recently been linked to mitigating plasticity loss by reducing covariate shift and promoting balanced neuron activations (Lyle et al., 2024a).

Unless otherwise specified, we benchmark these methods across three representative multi-task reinforcement learning algorithms: MTPPO, a shared-policy baseline with task-specific heads; a Mixture-of-Experts (MoE) model (Ceron et al., 2024); and MOORE (Hendawy et al., 2024), which incorporates orthogonal submodules for each task. This comprehensive evaluation allows us to disentangle the relative contributions of sparsity, explicit resets, and architectural regularization to plasticity preservation and multi-task performance. We report the normalized interquartile mean (IQM) with shaded regions indicating 95% stratified bootstrap confidence intervals, calculated using the `rliable` library (Agarwal et al., 2021).

**Environment and Benchmarks** We mostly consider the three multi-task MiniGrid (Chevalier-Boisvert et al., 2023) benchmarks proposed by Hendawy et al. (2024) – MT3, MT5, and MT7, with the exception being made for the results presented in Section 4.3, which use the MetaWorld MT10 benchmark (Yu et al., 2021). All environment details are outlined in Appendix B. We note that to ensure fair comparison across tasks with inherently different reward scales, for MiniGrid, raw episodic returns are normalized with respect to the maximum achievable reward in each environment (see Appendix B.3).

**Implementation and Training** For MiniGrid, we use the Proximal Policy Optimization (PPO) algorithm (Schulman et al., 2017) via the `mushroom_rl` library (D'Eramo et al., 2021) and the code provided by Hendawy et al. (2024) for multi-task architectures. Performance is measured by the episodic return across all tasks within the respective benchmark. Tasks are sampled randomly with replacement at the beginning of each episode during training. For MetaWorld, we use the Multi-Task Multi-Headed Soft Actor-Critic (MTMH SAC) (Haarnoja et al., 2018; Yu et al., 2021) and track the mean success rate across all tasks. We outline full training details and hyperparameters in Appendix A.

**Sparse Methods** To better characterize the role of sparsification in the MTRL setting, we began with a series of preliminary experiments comparing various sparsification strategies. Our goal was to identify methods that balance learning stability, generalization, and simplicity, while remaining compatible with the dynamic nature of multi-task settings. We evaluated several sparsification techniques, including: the Gradual Magnitude Pruning (GMP) schedule proposed by Zhu & Gupta (2017), Sparse Evolutionary Training (SET) (Mocanu et al., 2018), and Lottery Ticket Hypothesis (LTH) style rewinding (Frankle & Carbin, 2019). These initial experiments were conducted on the MT5 benchmark, selected as a practical compromise: it is more challenging than MT3, allowing us to meaningfully stress-test pruning strategies, yet significantly more computationally efficient than MT7, enabling extensive ablations at a reasonable cost. The results of this comparison are presented in Figure 1.

Among the three tested approaches, both GMP and SET resulted in more stable learning dynamics and improved generalization performance. As shown in Figure 1, sparse models trained with GMP and SET not only surpass their dense counterparts but also outperform single-task baselines. Conversely, LTH-based models fail to yield significant improvements over single-task training, highlighting their limited capacity to adapt in multi-task settings. We note that these results are consistent with findings in single-task reinforcement learning (Graesser et al., 2022; Obando-Ceron et al., 2024), and further underscore the advantages of sparsity mechanisms that adapt progressively throughout training. Given these insights, the remainder of our experimental study focuses on the two approaches that consistently demonstrated better performance: GMP and SET. The first, GMP, incrementally increases the network's sparsity level during training by gradually removing low-magnitude weights over a predefined time window. This allows the network to adapt to the increasing sparsity and mitigates the risk of destabilizing learning dynamics. For further information about the pruning schedule, we refer the reader to Appendix C.4. The second, SET, takes inspiration from evolutionary algorithms and maintains a fixed overall sparsity throughout training by continuously rewiring the network's connectivity. Unlike gradual pruning, which increases sparsity over time, SET preserves a

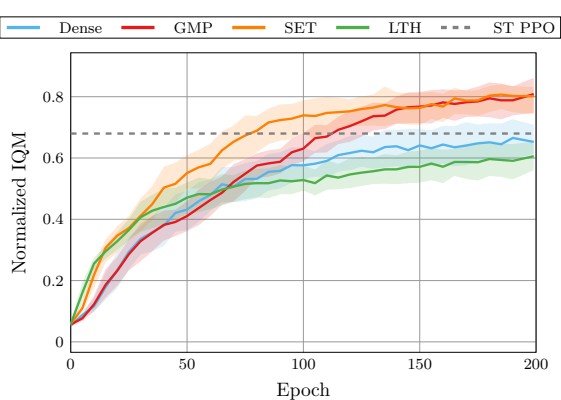

Figure 1: Comparative performance of various sparsification strategies (GMP, SET, LTH) against a dense multi-task baseline and an aggregation of single-task agents trained individually on each task on the MT5 benchmark. Both GMP and SET outperform the dense multi-task baseline and the LTH-based models, while LTH-based models show limited improvement over single-task performance and struggle to adapt effectively in the multi-task setting.

constant sparsity level but introduces dynamic plasticity through periodic topological updates. More details about the SET algorithm are presented in Appendix C.5. We note that both strategies are well aligned with the core objective of this work, which is to investigate the learning dynamics of sparse agents rather than to optimize inference-time performance. As both SET and GMP rely on unstructured pruning, they provide greater representational flexibility than structured approaches, making them particularly suitable for analyzing adaptation and interference in multi-task reinforcement learning (Hoefler et al., 2021).

**Plasticity Measures** We monitor three metrics during training, interpreted as correlative indicators of plasticity based on recent surveys and analyses (Berariu et al., 2023; Lyle et al., 2023; Klein et al., 2024; Falzari & Sabatelli, 2025), namely dormant neuron percentage, effective rank, and the trace of the Fisher Information Matrix. The computation of these metrics is detailed in Appendix C. Our analysis focuses on observing consistent patterns between pruning interventions, changes in these metrics, and MTRL performance, rather than claiming direct causality.

## 4 Core Effects of Sparse Methods

This section details our first set of empirical findings, beginning with the effect of sparsity on multi-task performance, followed by an analysis of its effects on plasticity indicators. All performance comparisons refer

to the aggregated final outcomes presented in Table 1. Plasticity metric analyses are primarily illustrated using the MT5 benchmark data shown in Figure 2, Figure 4, and Figure 5, as trends were generally consistent across other benchmarks unless otherwise stated. Detailed learning curves and plasticity metrics for all benchmarks are available in Appendix F and Appendix G, respectively.

Table 1: Final aggregate performance at epoch 200 across architectures and agent treatments on MT3, MT5, and MT7 benchmarks. **Gold** marks the best performance within the respective multi-task architecture and benchmark, while **blue** marks the second-best performance within each architecture and benchmark. Full learning curves illustrating training progression are available in Appendix F.

| Agent Treatment | MT3 | | MT5 | | MT7 | |
| --- | --- | --- | --- | --- | --- | --- |
| | IQM (↑) | 95% CI | IQM (↑) | 95% CI | IQM (↑) | 95% CI |
| *Multi-Task PPO (MTPPO)* | | | | | | |
| Dense | 0.70 | (0.60, 0.76) | 0.65 | (0.61, 0.71) | 0.72 | (0.69, 0.75) |
| Gradual Pruning | 0.77 | (0.73, 0.80) | 0.81 | (0.75, 0.86) | 0.76 | (0.70, 0.80) |
| SET | 0.76 | (0.74, 0.77) | 0.80 | (0.75, 0.84) | 0.80 | (0.77, 0.84) |
| ReDo | 0.74 | (0.68, 0.77) | 0.83 | (0.78, 0.84) | 0.80 | (0.76, 0.83) |
| Reset | 0.70 | (0.64, 0.72) | 0.80 | (0.74, 0.84) | 0.80 | (0.76, 0.83) |
| Weight Decay | 0.74 | (0.70, 0.78) | 0.75 | (0.66, 0.82) | 0.74 | (0.70, 0.77) |
| LayerNorm | 0.28 | (0.20, 0.37) | 0.33 | (0.25, 0.40) | 0.38 | (0.33, 0.45) |
| *Mixture of Experts (MoE)* | | | | | | |
| Dense | 0.74 | (0.71, 0.76) | 0.77 | (0.70, 0.82) | 0.80 | (0.75, 0.84) |
| Gradual Pruning | 0.77 | (0.74, 0.79) | 0.84 | (0.78, 0.86) | 0.87 | (0.83, 0.88) |
| SET | 0.76 | (0.74, 0.78) | 0.79 | (0.72, 0.85) | 0.82 | (0.78, 0.85) |
| ReDo | 0.77 | (0.76, 0.80) | 0.82 | (0.81, 0.85) | 0.85 | (0.82, 0.88) |
| Reset | 0.64 | (0.54, 0.73) | 0.78 | (0.73, 0.83) | 0.84 | (0.80, 0.87) |
| Weight Decay | 0.75 | (0.71, 0.76) | 0.78 | (0.70, 0.85) | 0.77 | (0.71, 0.82) |
| LayerNorm | 0.44 | (0.38, 0.46) | 0.34 | (0.29, 0.40) | 0.39 | (0.32, 0.43) |
| *Mixture of Orthogonal Experts (MOORE)* | | | | | | |
| Dense | 0.78 | (0.71, 0.80) | 0.84 | (0.81, 0.85) | 0.87 | (0.84, 0.88) |
| Gradual Pruning | 0.80 | (0.77, 0.81) | 0.85 | (0.80, 0.87) | 0.88 | (0.86, 0.89) |
| SET | 0.69 | (0.60, 0.74) | 0.78 | (0.72, 0.83) | 0.82 | (0.79, 0.85) |
| ReDo | 0.72 | (0.68, 0.75) | 0.82 | (0.78, 0.84) | 0.85 | (0.84, 0.87) |
| Reset | 0.67 | (0.62, 0.72) | 0.75 | (0.71, 0.79) | 0.84 | (0.80, 0.87) |
| WD | 0.75 | (0.73, 0.76) | 0.82 | (0.76, 0.87) | 0.87 | (0.84, 0.88) |
| LayerNorm | 0.54 | (0.49, 0.62) | 0.60 | (0.53, 0.65) | 0.66 | (0.61, 0.70) |

## 4.1 Sparse Methods Improve Task Performance

Our findings indicate that both GMP and SET generally lead to improvements in multi-task performance, an observation consistent with a significant body of research in supervised learning, where appropriately pruned sparse networks have been shown to match, outperform, and generalize better than their dense counterparts (Guo et al., 2019; Morcos et al., 2019; Sabatelli et al., 2020; Hoefler et al., 2021). However, in our multi-task settings, the extent of these benefits from pruning varies with the underlying agent architecture and desired sparsity level. For MTPPO and MoE architectures, both Gradual Pruning and SET consistently resulted in improved final aggregate returns compared to their respective dense baselines across all tested benchmarks (MT3, MT5, MT7), as shown in Table 1. This suggests that these common MTRL architectures frequently contain considerable overparameterization that sparse methods can effectively address, hinting at a direct link between sparse intervention and improved MTRL outcomes. In contrast, the impact of sparse methods on MOORE was more nuanced. While in general, the effect of GMP on MOORE on performance was close to that of the dense baseline, SET led to a slight decline across all benchmarks. While substantial gains were not observed for MOORE with sparsification, the ability to prune to high levels of sparsity (up to

95%) without significant performance degradation still indicates that even sophisticated architectures can be overparameterized. Nonetheless, we note that very aggressive pruning (e.g., 99% sparsity with GMP) could lead to issues such as rank collapse or performance drops in MOORE (see Appendix H, Figure 23 and Figure 24).

## 4.2 Sparse Methods Mitigate Plasticity Loss

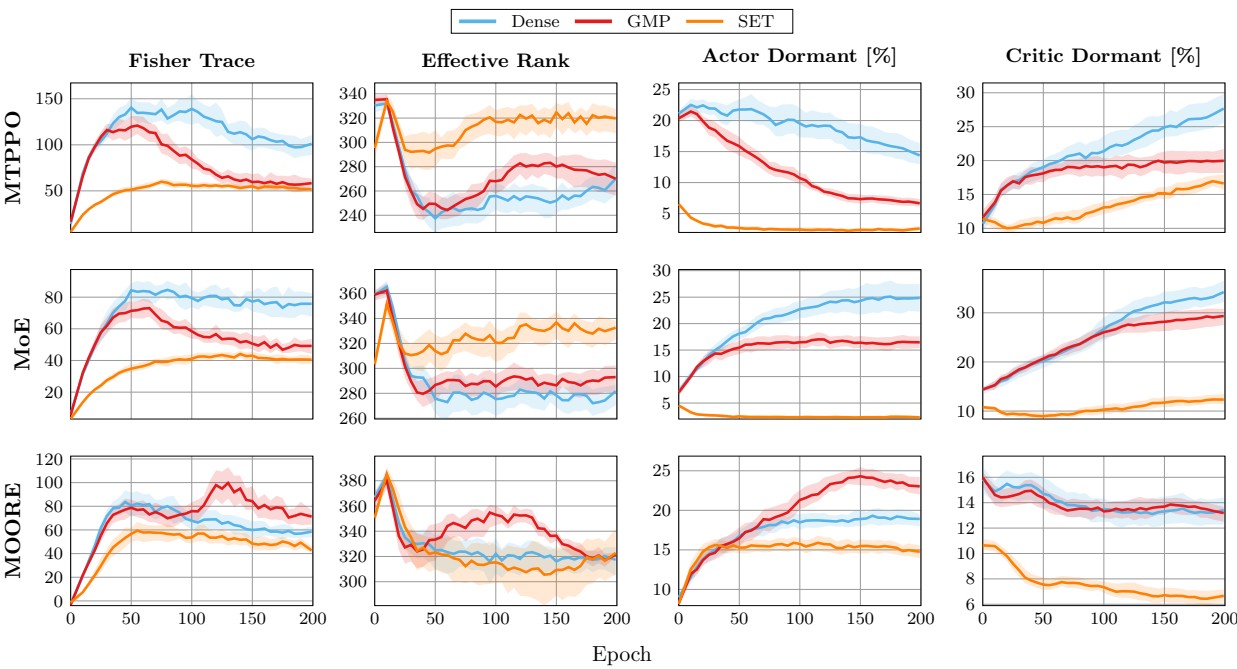

Figure 2: Evolution of plasticity indicators for Dense, Gradual Magnitude Pruning (GMP), and Sparse Evolutionary Training (SET) across different MTRL architectures (MTPPO, MoE, MOORE) on the MT5 benchmark. Subplots display Fisher Trace, Effective Rank, and percentages of Actor and Critic Dormant neurons, illustrating the distinct effects of each sparsification strategy compared to the dense baseline.

The observed performance improvements, particularly within the MTPPO and MoE architectures, strongly correlate with the ability of sparsification methods to mitigate common indicators of plasticity loss, while displaying distinct learning dynamics for GMP and SET. Notably, the plasticity profiles (Figure 2) show that agents employing either GMP or SET generally exhibit more favorable plasticity metrics compared to their dense counterparts. Specifically, sparse agents typically maintain lower percentages of dormant neurons and a higher or more stable mean effective rank in their representations. Furthermore, the trace of the Fisher Information Matrix (FIM) in sparse agents typically stabilizes at lower values post-initial learning, suggesting convergence to less sensitive parameter configurations, in contrast to the often persistently high values in dense networks. While both GMP and SET contribute to these general improvements over dense networks, they induce individually different plasticity dynamics. SET, with its continuous rewiring, proved particularly effective at minimizing neuron dormancy to very low levels in both actor and critic components throughout training, while also maintaining a higher effective rank. In contrast, GMP's impact on dormancy was more pronounced in the actor network, with both actor and critic dormant percentages remaining higher than those under SET, though still an improvement over dense networks. The FIM trace also differed: GMP often displayed a characteristic peak-and-decline pattern, whereas SET maintained a low and stable FIM trace throughout training, suggesting continuous adaptation within a less volatile optimization regime. Collectively, these observations support the hypothesis that sparsification methods enhance the learning capability of MTRL agents, plausibly through the mitigation of processes associated with plasticity degradation in dense networks. Nevertheless, the influence of sparsification on MOORE's plasticity did not mirror the benefits seen in MTPPO and MoE agents, aligning with the more varied overall performance outcomes discussed

above. For MOORE, SET did reduce neuron dormancy, and its Fisher Trace showed a slow growth and stabilization pattern. However, the effective rank for both SET and GMP remained similar to the dense baseline. GMP, in contrast to its effect in other architectures, sometimes even slightly increased dormancy compared to dense MOORE on certain metrics. Importantly, these specific plasticity modulations, such as SET's reduced dormancy in MOORE, generally did not translate into performance improvements for this architecture, with SET often resulting in a slight performance decline. This suggests that MOORE's inherent design, particularly its emphasis on representation orthogonalization (Hendawy et al., 2024), may interact with sparsification in various ways. Its sophisticated structure might be less responsive to the typical benefits derived from these plasticity changes, as it already exhibits relatively stable plasticity characteristics.

### 4.3 Generalization to Continuous Control

To evaluate whether our plasticity-related findings in MiniGrid generalize to continuous control, we extended our analysis to the MetaWorld MT10 benchmark (Yu et al., 2021).

Our experimental setup was guided by two insights from McLean et al. (2025): increasing the critic's capacity tends to yield greater benefits than increasing the actor's, making the critic the more capacity-sensitive component; and while overall plasticity loss (e.g., neuron dormancy) is relatively low in dense agents, it tends to be more pronounced in the actor than in the critic. These findings led us to hypothesize that pruning only the actor, while preserving the full capacity of the critic, could enhance performance by improving network efficiency without compromising representational power. We evaluated this hypothesis by comparing three conditions: a dense MTMH-SAC baseline, GMP applied to both actor and critic, and GMP applied to the actor only. The actor-only pruning approach achieved the highest final success rate at 81% (95% CI: 0.77, 0.83), outperforming both the dense baseline (73%; 95% CI: 0.67, 0.75) and the global pruning condition (75%; 95% CI: 0.73, 0.78); see Appendix F, Figure 13 for exact learning curves. This performance improvement was accompanied by a sustained reduction in actor neuron dormancy, as depicted in Figure 3, suggesting a more adaptive and efficient use of network capacity. Overall, these results extend our MiniGrid-based plasticity findings to the more complex MetaWorld benchmark and offer

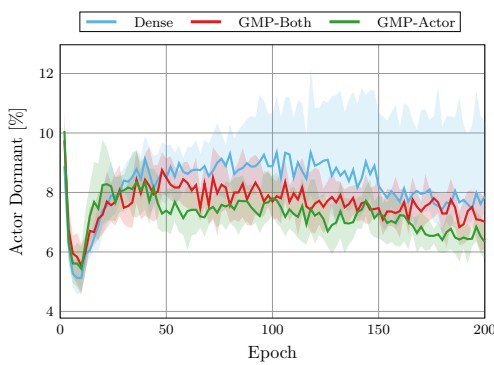

Figure 3: Reduction in actor neuron dormancy on MetaWorld MT10. Selective pruning of the actor network (GMP-Actor) leads to a sustained decrease in dormant neurons compared to both the dense MTMH-SAC baseline and the globally pruned model (GMP-Both).

a complementary perspective to McLean et al. (2025): while they emphasize scaling the critic, we show that selectively pruning the actor can be equally beneficial. Together, these insights highlight the asymmetry in actor-critic dynamics and suggest that the benefits of sparsity are both role and context-dependent.

## 5 Interactions with Alternative Mechanisms

This section shifts focus to a comparative analysis between sparsification and other alternative strategies for plasticity and multi-task learning. We first examine pruning in relation to explicit interventions such as ReDo and Reset, which directly manipulate network parameters to counteract plasticity loss. We then consider implicit mechanisms such as standard regularization techniques (Weight Decay) and architectural choices (LayerNorm) that influence plasticity without explicit intervention. All comparisons are conducted under the same multi-task training setup and are summarized in Table 1. Additionally, we present a final ablation study exploring potential synergies of combining GMP with other optimization techniques (Weight Decay and PCGrad (Yu et al., 2020)).

## 5.1 Sparsification versus Explicit Plasticity-Inducing Mechanisms

We compared GMP and SET against interventions that explicitly target symptoms of plasticity loss: ReDo (reinitializing dormant neurons) and Reset (layer reinitialization), using their best-performing configurations derived after hyperparameter tuning (see Figure 11 of Appendix E). In terms of final task performance (Table 1), sparsification methods generally achieved returns competitive with, and occasionally better than, ReDo or Reset, especially for MTPPO and MoE architectures. While statistical significance for outperformance was not always established due to overlapping confidence intervals, sparse methods consistently presented a strong alternative without directly targeting specific plasticity symptoms. For MOORE agents, performance differences between ReDo and the sparse approaches were minimal, while Reset introduced substantial variability. Examining the plasticity profiles (Figure 4), SET was particularly effective for MTPPO and MoE, often maintaining a lower percentage of actor dormant neurons than even ReDo and consistently achieving the highest effective rank. In contrast, within MOORE, ReDo was more effective in reducing dormancy, while SET's effective rank advantage was less apparent. ReDo's impact on the Fisher Trace and mean effective rank often mirrored that of the dense baseline, indicating it primarily addressed dormancy without broadly altering other representational characteristics. The Reset intervention, due to its periodic reinitializations, frequently induced abrupt shifts and instability in markers like the FIM and effective rank, especially post-reset, consistent with prior work (Falzari & Sabatelli, 2025). Performance-wise, Reset rarely outperformed sparse agents, whereas ReDo was more competitive; however, achieving a lower percentage of dormant neurons via ReDo did not always guarantee superior task performance (e.g., in MOORE), and SET sometimes achieved lower actor dormancy without this direct targeting.

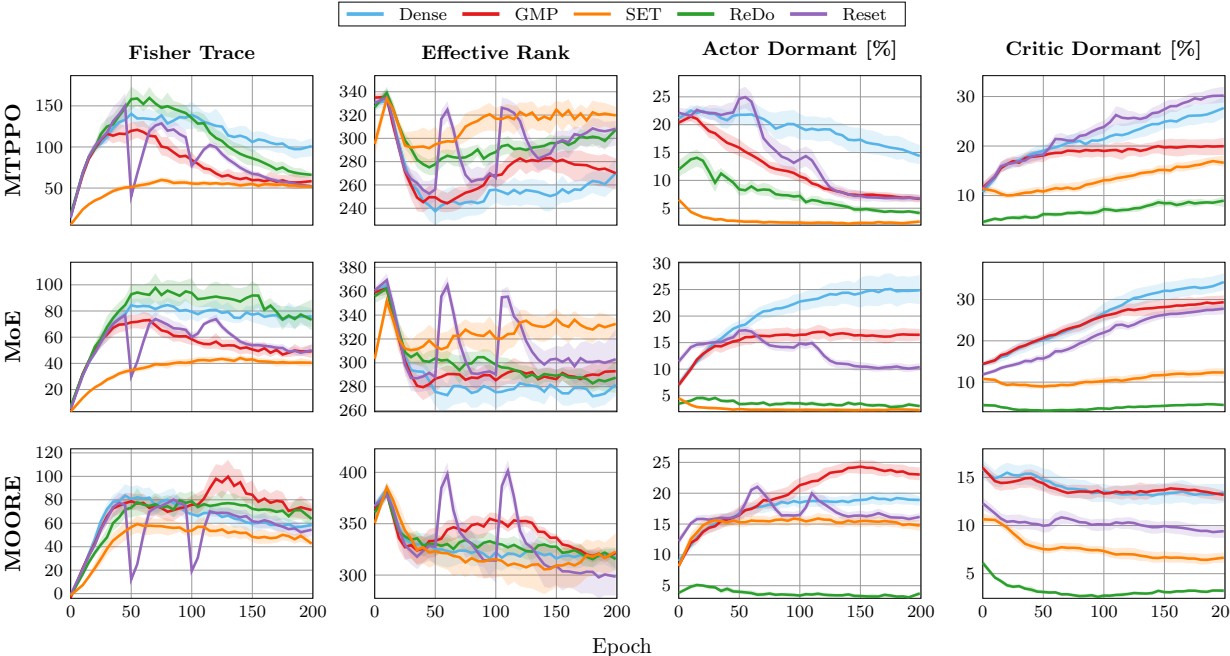

Figure 4: Comparative plasticity dynamics of sparse methods (GMP, SET) versus explicit plasticity-inducing interventions (ReDo, Reset) and a dense baseline, across MTPPO, MoE, and MOORE architectures on the MT5 benchmark. Metrics include Fisher Trace, Effective Rank, and percentage of Actor and Critic Dormant Neurons.

## 5.2 Sparsification versus Implicit Plasticity-Inducing Mechanisms

To further characterize the role of sparsification in fostering plasticity, we contrast its effects with more implicit plasticity-inducing mechanisms: weight regularization (Weight Decay, WD) and architectural normalization (LayerNorm). These techniques have been explored for mitigating plasticity loss by promoting parameter

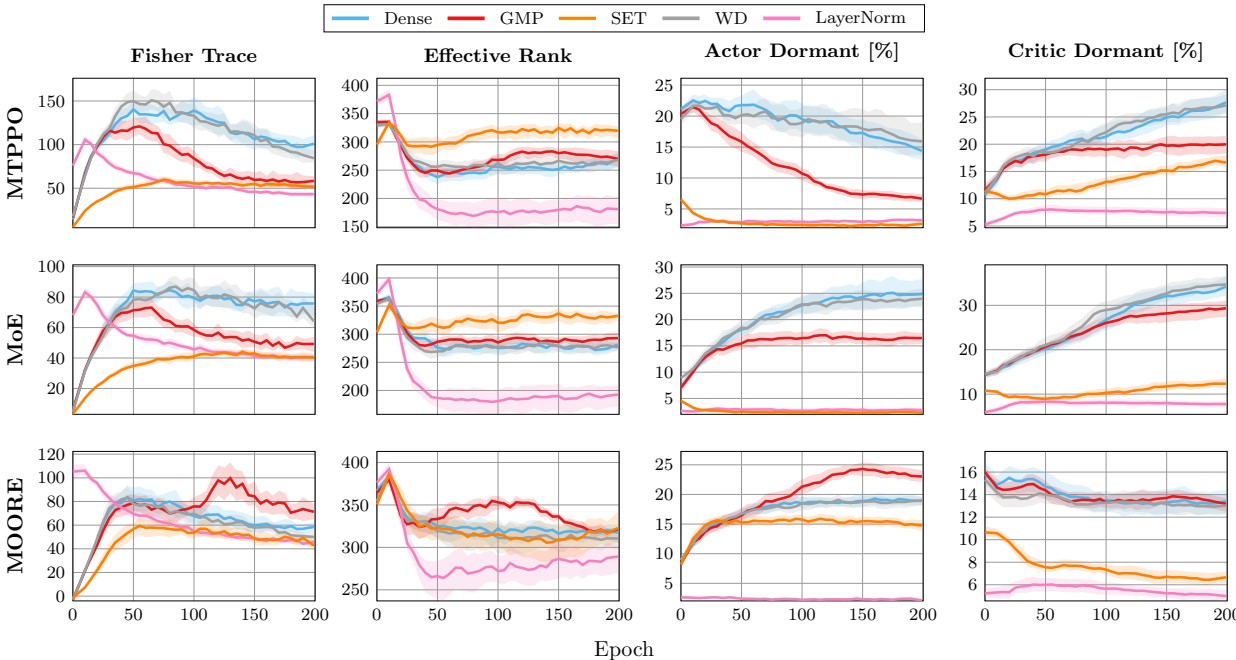

Figure 5: Comparative plasticity dynamics of sparse methods (GMP, SET) versus regularization techniques (Weight Decay, LayerNorm) and a dense baseline, across MTPPO, MoE, and MOORE architectures on the MT5 benchmark. Metrics include Fisher Trace, Effective Rank, and percentage of Actor and Critic Dormant Neurons.

stability or scale-invariant updates (Lyle et al., 2023). These experiments evaluate whether sparsification methods like GMP and SET offer distinct advantages over, or complementary benefits to, these common regularization approaches. In terms of task performance, both GMP and SET generally outperform agents trained with only Weight Decay or LayerNorm across all architectures and benchmarks. In Figure 5, Weight Decay exhibits plasticity dynamics remarkably similar to the dense baseline across all metrics and architectures. This suggests that while WD is a common regularizer, in these MTRL contexts, it does not substantially alter plasticity characteristics beyond a standard dense network, nor does it typically lead to performance surpassing well-configured sparse agents. LayerNorm, conversely, induces more pronounced changes to plasticity. Even though it can lead to very low levels of dormant neurons, this is accompanied by a severe and persistent reduction in the effective rank. This drop, visible across all architectures, signifies limited representational diversity, which also correlates with its lowered performance. We hypothesize that in the multi-task learning setting, LayerNorm, by normalizing activations across features within each layer and sample, might inadvertently introduce strong correlations in the gradients from different tasks or smooth out task-specific feature distinctions excessively. This could lead to a less expressive representation space, hindering the network's ability to learn diverse task solutions despite the apparent reduction in neuron dormancy. The low effective rank, coupled with often the atypically low Fisher Traces for a dense agent, likely contributes to LayerNorm's consistently poor task performance. In contrast, sparsification methods like GMP and SET generally achieve a better balance: they effectively mitigate dormancy (SET often being most effective) and maintain or improve effective rank compared to the dense baseline (especially GMP initially), without the severe representational collapse seen with LayerNorm. This suggests that sparsification offers a more nuanced approach to capacity control and plasticity preservation than these standard implicit regularization techniques in the studied MTRL scenarios, leading to superior overall learning outcomes.

## 5.3 Interaction with Optimizers

We finally explored the potential synergies of combining GMP with other optimization techniques, specifically weight decay and PCGrad (Yu et al., 2020), a popular method designed to mitigate

gradient interference in multi-task settings, on the MTPPO architecture using the MT5 benchmark. The performance results are shown in Figure 6 and plasticity dynamics in Figure 7. In terms of final task performance, GMP in isolation achieved the highest returns (Figure 6).

While PCGrad alone improved performance over the dense baseline, its combination with GMP did not yield further gains beyond GMP alone. Interestingly, the plasticity profiles also reveal that GMP alone maintained the most favorable characteristics (Figure 7). The GMP+WD combination showed similar plasticity dynamics to GMP alone, although with slightly worse values in some metrics, correlating with its slightly lower performance. This lack of synergy with Weight Decay might be anticipated, as WD encourages smaller weight magnitudes overall, potentially increasing the pool of weights that magnitude-based pruning would target, which could lead to a less discerning pruning process or even premature removal of weights that might have otherwise become important. PCGrad, when applied to a dense network, did not demonstrably improve plasticity indicators such as Fisher Trace or actor dormancy compared to the dense baseline, despite its performance uplift. This suggests that the primary benefits of GMP in this context may stem from its inherent regularization effects and capacity optimization, which are not necessarily enhanced by, or may even be slightly counteracted by, the addition of these particular optimizers.

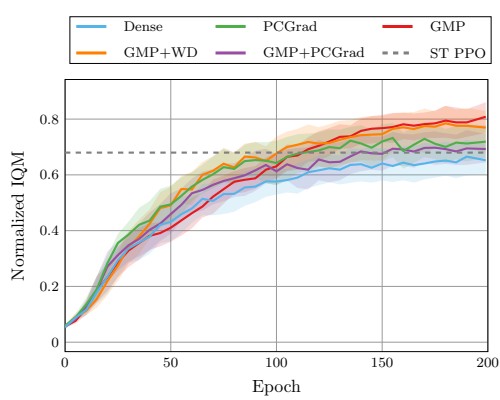

Figure 6: Performance comparison for Gradual Magnitude Pruning (GMP) interactions with Weight Decay (WD) and PCGrad on the MTPPO architecture with the MT5 benchmark.

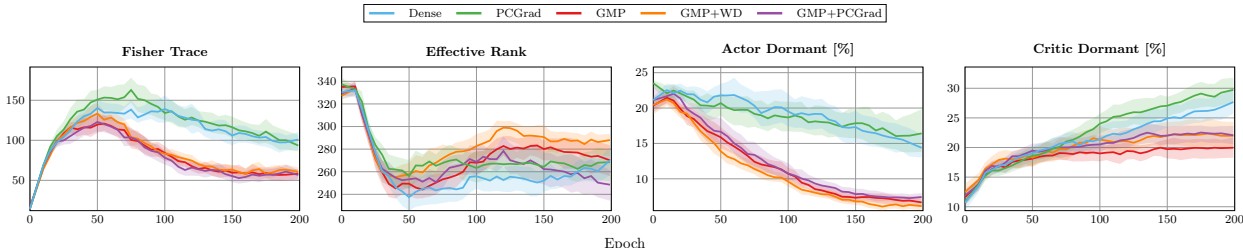

Figure 7: Comparative plasticity dynamics for Gradual Magnitude Pruning (GMP) and its interactions with Weight Decay (GMP+WD) and PCGrad (GMP+PCGrad), contrasted with PCGrad on a dense network and a dense baseline. All experiments are on the MTPPO architecture with the MT5 benchmark. Subplots display (from left to right): Fisher Trace, Effective Rank, Actor Dormant %, and Critic Dormant %.

## 6 Considerations of Sparse Methods

While this study is primarily empirical, the observed benefits of sparsification methods in MTRL can be interpreted through several established concepts from the sparsity, optimization, and multi-task learning literature. However, we also want to acknowledge their limitations and considerations for practical application.

**Optimization, Sparsity, and Generalization** The iterative removal (GMP) or rewiring of connections (SET) effectively guides the network towards sparse solutions. This mechanism can generally be viewed as $L_0$ regularization, encouraging sparsity by penalizing the number of non-zero parameters (Louizos et al., 2018) and reducing the model's degrees of freedom. While this might confine optimization to lower-dimensional subspaces (Gao & Jojic, 2016; Hoefler et al., 2021), such sparse solutions are often associated with "flatter" minima in the loss landscape (Peste, 2023; Shah et al., 2024). Flatter minima are highly desirable due to lower sensitivity to parameter perturbations (Foret et al., 2021; Lee et al., 2025b), widely believed to result in better generalization and robustness under distribution shifts (Hochreiter & Schmidhuber, 1997; Jiang et al., 2019; Kaddour et al., 2023; Li et al., 2024), the primary drivers of plasticity loss. The convergence to such local flat minima is often indicated by specific dynamics in the curvature of the loss for instance, the

maximal Hessian eigenvalue typically grows, peaks, and then declines during training (Fort & Ganguli, 2019). Our empirical results for MTPPO and MoE, where the Fisher Trace (a proxy for curvature (Lewandowski et al., 2024b)) exhibited a peak-and-decline pattern with GMP (Figure 2), align with convergence to such flatter minima. SET, with its continuous rewiring, maintained a low, stable Fisher trace, suggesting a robust optimization process.

This aligns with findings in supervised learning where sparse networks are recognized for reduced overfitting and better generalization than dense counterparts (Gopalakrishnan et al., 2018; Cosentino et al., 2019; Guo et al., 2019; Liu et al., 2019; Liu, 2020). The iterative nature of GMP and the dynamic regrowing of SET are generally thought to help models evade suboptimal local minima (Jin et al., 2016; Gale et al., 2019; Hoefler et al., 2021; Jin et al., 2022; Graesser et al., 2022), a principle consistent with our findings. Furthermore, the success of pruning in MTRL models with shared backbones and task-specific heads mirrors similar effectiveness in multi-task supervised learning Xiang et al. (2024). Nonetheless, an excessive reduction in degrees of freedom via aggressive pruning can hinder the satisfaction of specific architectural demands, such as maintaining expert orthogonality in MOORE (see Appendix H, Figure 23 and Figure 24), especially if capacity becomes overly constrained.

**Limitations and Practical Considerations** Despite their benefits, these sparsification techniques have considerations. GMP, while conceptually simple, typically operates on dense weight matrices internally during training, applying masks to simulate sparsity. This means it does not inherently reduce the memory footprint or computational cost during training compared to dense models; true benefits often require specialized hardware or software for sparse operations at inference. While inference can be efficient, the training phase still bears the overhead of the original dense model, with the additional overhead of pruning at specified timesteps. SET, on the other hand, can maintain true sparsity throughout training and inference if implemented with sparse data structures. However, current widely available implementations are often optimized for fully connected layers, and extending their dynamic rewiring efficiently to convolutional or recurrent architectures can be more complex. Additionally, the random nature of SET's regrowth phase, while promoting exploration, might not always lead to the most optimal connectivity patterns without more guided heuristics. Both methods also require careful tuning of their own hyperparameters to achieve optimal results. While potentially less sensitive than some explicit plasticity interventions, this still constitutes a tuning effort. The architecture-dependent nature of the benefits, as shown by our experiments with MOORE, also indicates that these are not one-size-fits-all solutions.

## 7 Conclusion

In this work, we examined dynamic sparsification, specifically Gradual Magnitude Pruning (GMP) and Sparse Evolutionary Training (SET), as a means to mitigate plasticity loss and improve performance in multi-task reinforcement learning (MTRL). Our results show that both methods can enhance adaptability and generalization across several architectures, with consistent gains observed in MTPPO and MoE agents. Similar benefits were also found in Multi-Headed SAC agents evaluated on the MetaWorld MT10 benchmark, suggesting that the effectiveness of sparsification extends to continuous control tasks. While performance on MOORE was more variable, likely due to its built-in mechanisms for managing interference, our findings highlight the importance of aligning sparsification strategies with architectural design. In addition to performance improvements, sparsity can offer benefits such as reduced hyperparameter sensitivity, computational efficiency, and implicit regularization through structured parameter removal. These results support the view that general-purpose mechanisms that shape learning dynamics rather than task-specific interventions can yield robust benefits in MTRL. As future work, we plan on investigating the theoretical underpinnings and potential interpretability advantages of sparse MTRL models.

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

## A    Training Details and Hyperparameters

This appendix details the hyperparameters used for the experimental evaluations presented in this study. For MiniGrid, we report the IQM and CIs of the episodic return over 30 random seeds, whereas for MetaWorld, we report the IQM and CIs of the mean success rate across 10 seeds. Table 2 provides a list covering the general experimental settings, architecture of the used networks, and specific hyperparameters used for MoE and MOORE. All architectures are multi-headed, with task-specific heads. Hyperparameters were largely adopted from Hendawy et al. (2024), with modifications in the number of training epochs, number of evaluation episodes, and evaluation frequency. Table 3 outlines the training details and parameters for MetaWorld MT10. The full implementation is available at https://github.com/atodorov284/sparsity_driven_plasticity.

## B    Environment Details

This appendix provides details on the MiniGrid (Chevalier-Boisvert et al., 2023) environments used in our multi-task benchmarks. We use standard environments from the MiniGrid suite, which are designed to test various capabilities such as navigation, memory, and problem-solving in partially observable grid-world settings with sparse reward. For environmental details on MetaWorld, we refer the reader to Yu et al. (2021).

### B.1    Composition

Our experiments use three multi-task benchmarks – MT3, MT5, and MT7, as proposed by Hendawy et al. (2024), composed as follows:

- MT3: `LavaGapS7-v0` + `RedBlueDoors-6x6-v0` + `MemoryS11-v0`

- MT5: MT3 + `DoorKey-6x6-v0` + `DistShift1-v0`

- MT7: MT5 + `SimpleCrossingS9N2` + `MultiRoom-N2-S4`

### B.2    Descriptions

Below are descriptions for each unique environment used in the benchmarks, adapted from Chevalier-Boisvert et al. (2023). In all environments `S` specifies the size of the map `SxS`.

- `DoorKey-6x6-v0`: The agent must pick up a key, navigate to a locked door, and open it to reach a goal square.

- `DistShift1-v0`: The agent starts in the top-left corner and must reach the goal, which is in the top-right corner, but has to avoid stepping into lava on its way. Stepping into lava terminates the episode.

- `RedBlueDoors-6x6-v0`: The agent is in a room with two doors, one red and one blue. The agent has to open the red door and then open the blue door, in that order.

- `MemoryS11-v0`: The agent starts in a small room where it sees an object. It then has to go through a narrow hallway, which ends in a split. At each end of the split, there is an object, one of which is the same as the object in the starting room. The agent has to remember the initial object and go to the matching object at the split.

- `SimpleCrossingS9N2-v0`: The agent has to reach the green goal square on the other corner of the room while avoiding walls. Walls run across the room either horizontally or vertically, and have `N` crossing points which can be safely used; the path to the goal is guaranteed to exist.

- `MultiRoom-N2-S4-v0`: This environment has a series of connected rooms with doors that must be opened to get to the next room. The final room has the green goal square that the agent must get to. `N` specifies the number of rooms.

Table 2: Core experimental setup, agent architecture, and algorithm hyperparameters on MiniGrid. The choice for hyperparameters is largely borrowed from Hendawy et al. (2024), while following their exact training configuration (except number of evaluation episodes).

| Hyperparameter | Value |
|---|---|
| *General:* | |
| Number of environments | [3, 5, 7] |
| Steps per environment | 1 step per environment |
| Number of epochs | 200 |
| Steps per epoch | 2000 |
| Total number of timesteps | 400000 |
| Train frequency | 2000 timesteps |
| Evaluation episodes | 25 per task |
| Evaluation frequency | 10000 timesteps |
| *Shared Feature Extractor:* | |
| Type | Conv2D |
| Channels per Layer | [16, 32, 64] |
| Kernel Size | [(2,2), (2,2), (2,2)] |
| Activations | [ReLU, ReLU, Tanh] |
| *PPO:* | |
| Optimizer | Adam (Kingma & Ba, 2017) |
| Critic Loss | MSE |
| Actor Learning Rate | $1 \times 10^{-3}$ |
| Critic Learning Rate | $1 \times 10^{-3}$ |
| Critic Network Hidden Size | 128 |
| Actor Network Hidden Size | 128 |
| Number of Linear Layers | $2 \times |\mathcal{T}|$ (number of tasks) |
| Number of Output Units | $|\mathcal{A}|$ for actor, 1 for critic |
| Output Activations | [Tanh, Linear] |
| GAE $\lambda$ | 0.95 |
| Entropy Term Coefficient | 0.01 |
| Clipping $\varepsilon$ | 0.2 |
| Epochs for Policy | 8 |
| Epochs for Critic | 1 |
| Batch Size for Policy | 256 |
| Batch Size for Critic | 2000 |
| Discount Factor ($\gamma$) | 0.99 |
| *Task Encoder (for MoE/MOORE):* | |
| $k$ Experts | [2, 3, 4] |
| Encoder Linear Layers | 1 |
| Encoder Output Units | $k$ (number of experts) |
| Encoder Use Bias | False |
| Encoder Activation | Linear |

Table 3: The hyperparameters and training setup used for MTMH SAC on MetaWorld MT10.

| Hyperparameter | Value |
|---|---|
| *General Training:* | |
| Total Timesteps | 20000000 |
| Batch Size | 1280 |
| Replay Buffer Size | 1000000 |
| Warmstart Steps | 40000 |
| Evaluation Frequency | 200000 steps |
| Number of Epochs | 200 |
| Number of Updates | 2000000 |
| Number of Tasks | 10 |
| Evaluation Episodes | 50 per task |
| Max Episode Steps | 500 |
| *SAC:* | |
| Discount Factor ($\gamma$) | 0.99 |
| Target Smoothing Coeff. ($\tau$) | 0.005 |
| Number of Critics | 2 |
| Initial Temperature ($\alpha$) | 1.0 |
| Target Q-Value Clip | 5000 |
| *Actor and Critic* | |
| Optimizer | Adam |
| Layer Type | Linear |
| Learning Rate | $3 \times 10^{-4}$ |
| Max Gradient Norm | 1.0 |
| Network Depth | 3 |
| Hidden Size | 400 |
| Activation | ReLU |
| Log-Std Bounds | $[-20, 2]$ |
| *Temperature Optimizer:* | |
| Optimizer | Adam |
| Learning Rate | $3 \times 10^{-4}$ |
| Max Gradient Norm | None |
| *Gradual Magnitude Pruning:* | |
| Desired Sparsity $\rho_F$ | 95% |
| Pruning Frequency $f_p$ | 5000 timesteps |
| Pruning start interval $t_{\text{start}}$ | $0.05\times$ number of timesteps |
| Pruning end interval $t_{\text{end}}$ | $0.80\times$ number of timesteps |
| Sparsity $\rho_t$ at timestep $t$ | $\rho_F \left[ 1 - \left( 1 - \frac{t - t_{\text{start}}}{t_{\text{end}} - t_{\text{start}}} \right)^3 \right]$ |

- `LavaGapS7-v0`: The agent has to reach the green goal square at the opposite corner of the room, and must pass through a narrow gap in a vertical strip of deadly lava. Touching the lava terminates the episode with a zero reward.

### B.3 Reward Normalization

To ensure fair comparison across tasks with inherently different reward scales, raw episodic returns are normalized with respect to the maximum achievable reward in each environment. The standard MiniGrid reward for successful task completion is calculated as

$$1 - 0.9 \times (\texttt{steps\_taken}/\texttt{max\_episode\_steps}),$$

while failure results in a score of 0. We further normalize this score by performing a Min-Max scaling with respect to the maximum performance obtainable in each environment. We use the default maximum timesteps of each environment and estimate how many steps an optimal agent can solve the environment. This normalization procedure scales the performance such that a score of 1.0 represents achieving the optimal (shortest path) solution, facilitating comparisons of learning efficacy across environments with varying complexities and step horizons and addressing reward scales. Table 4 presents the optimal steps, maximum allowed steps, and the maximum achievable reward used for the score normalization of each environment.

Table 4: Environment-specific parameters for reward normalization. This table lists the optimal number of steps to solve each task, the maximum default permissible steps per episode, and the resulting maximum achievable raw reward score (used as the max score for normalization).

| Environment Name | Optimal Steps | Max Steps | Achievable Reward |
|---|---|---|---|
| `DoorKey-6x6-v0` | 11 | 360 | 0.9725 |
| `DistShift1-v0` | 11 | 252 | 0.9607 |
| `RedBlueDoors-6x6-v0` | 8 | 720 | 0.9900 |
| `LavaGapS7-v0` | 8 | 196 | 0.9633 |
| `MemoryS11-v0` | 15 | 605 | 0.9777 |
| `SimpleCrossingS9N2-v0` | 15 | 324 | 0.9583 |
| `MultiRoom-N2-S4-v0` | 5 | 40 | 0.8875 |

## C Implementation Details

This appendix details the methodology and interpretation of the plasticity metrics, pruning schedule, and the sparse evolutionary training used in this work. The plasticity measures serve as correlative indicators of an agent's learning capacity and adaptability. The computation of activations and gradients for these metrics relies on sampling from a plasticity replay buffer of training observations to approximate expected values via sample means. Hyperparameters specific to these calculations are detailed in Table 5.

### C.1 Neuron Dormancy

We adapt the dormant neuron formalization from Sokar et al. (2023). A neuron's activity is assessed relative to other (non-masked) neurons in the same layer. Given an input distribution $D$ (approximated by the plasticity replay buffer) and an activation $h_i^l(x)$ of a neuron $i$ in layer $l$ with $H^l$ neurons under input $x \in D$, the normalized activation is

$$s_i^l = \frac{\mathbb{E}_{x \in D}|h_i^l(x)|}{\frac{1}{H^l} \sum_{k=1}^{H^l} \mathbb{E}_{x \in D}|h_k^l(x)|}.$$

Neuron $i$ is called $\tau$-dormant for some threshold $\tau > 0$ if $s_i^l \leq \tau$. If $H_\tau^l$ denotes the number of dormant neurons per layer, then the *dormancy ratio* $\beta_\tau$ is the ratio of dormant neurons and all neurons across all

layers in the network $L_{\text{all}}$ except the final $L_{\text{out}}$

$$\beta_\tau = \frac{\sum_{l \in L_{\text{all}} \setminus \{L_{\text{out}}\}} H_\tau^l}{\sum_{l \in L_{\text{all}} \setminus \{L_{\text{out}}\}} H^l}.$$

A high percentage of dormant neurons suggests significant underutilization of the network's capacity, potentially hindering its ability to learn complex functions or adapt to new data, a key aspect of plasticity. For the ReDo method, dormant neurons were reinitialized at specific time intervals $f_d$ and a threshold $\tau$, determined through the hyperparameter sweeps in Appendix E

## C.2  Trace of the Fisher Information Matrix

The Fisher Information Matrix (FIM) $F$ quantifies the sensitivity of a model's output (e.g., the policy) to changes in the parameters $\theta$ (Klein et al., 2024; Falzari & Sabatelli, 2025; Ven, 2025). For a policy $\pi$, its Fisher trace is given by

$$\text{Tr}(F) = \mathbb{E}_{s,a \sim \pi} \left[ \|\nabla_\theta \log \pi(a|s)\|_2^2 \right].$$

The trace of the FIM can be viewed as a measure of the policy's sensitivity to parameter perturbations. A very high or persistently increasing trace might indicate that the policy is in a "sharp" region of the loss landscape, making it brittle to small changes and potentially indicative of overfitting or optimization instability. Conversely, a lower, stabilized trace, as observed in our pruned agents (see Section 4.2), can suggest convergence to "flatter" minima, implying a more robust policy that is less sensitive to parameter variations and more capable of sustained learning or adaptation.

## C.3  Effective Rank

For a feature matrix $\Phi$ (e.g., a shared feature extractor) with $d$ singular values $\sigma_i$ sorted descendingly, the effective rank at tolerance $\delta$ is

$$\text{srank}_\delta(\Phi) = \min_k \left\{ \frac{\sum_{i=1}^k \sigma_i}{\sum_{i=1}^d \sigma_i} \geq 1 - \delta \right\}.$$

The effective rank measures the dimensionality of the space spanned by the features. A low effective rank suggests a representation collapse, where learned features are highly correlated and less diverse, limiting a network's ability to represent various information. Conversely, a high effective rank implies a richer, more diverse set of feature representations, implying a greater capacity to learn and distinguish between inputs.

## C.4  Gradual Magnitude Pruning (GMP)

We implement the pruning schedule proposed by Zhu & Gupta (2017), which progressively increases network sparsity during training. At regular pruning intervals (defined by a pruning frequency $f_p$), connections (weights) with the smallest absolute magnitudes are masked (set to zero). This process continues until a target sparsity level $\rho_t$ is achieved for the current training step $t$. The sparsity level $\rho_t$ follows a cubic growth schedule, gradually increasing from an initial sparsity at $t_{\text{start}}$ to a final target sparsity $\rho_F$ at $t_{\text{end}}$:

$$\rho_t = \begin{cases} 0 & \text{if } t < t_{\text{start}}, \\ \rho_F \left[ 1 - \left( 1 - \frac{t - t_{\text{start}}}{t_{\text{end}} - t_{\text{start}}} \right)^3 \right] & \text{if } t_{\text{start}} \leq t \leq t_{\text{end}}, \\ \rho_F & \text{if } t > t_{\text{end}}. \end{cases}$$

This schedule allows the network to adapt to increasing levels of sparsity rather than undergoing abrupt structural changes. The specific values for $\rho_F$, pruning frequency, $t_{\text{start}}$, and $t_{\text{end}}$ used in our experiments were determined through ablation studies (see Appendix D).

## C.5 Sparse Evolutionary Training (SET)

The Sparse Evolutionary Training (SET) mechanism, inspired by Mocanu et al. (2018), maintains a constant overall network sparsity $\rho = 1 - (\|\mathbf{W}\|_0 / N_{\text{total}})$ throughout training, where $\|\mathbf{W}\|_0$ is the total number of non-zero weights and $N_{\text{total}}$ is the total number of parameters in the sparsified layers. At predefined evolution intervals, a fraction $\varepsilon$ of the existing connections with the smallest absolute magnitudes $|w_{ij}|$ are pruned. To preserve the sparsity level $s$, an equivalent number of new connections, $N_{\text{new}} = \varepsilon \cdot \|\mathbf{W}\|_0^{\text{current}}$, are simultaneously regrown. These new connections are typically introduced randomly at locations within the network that currently have zero weight, allowing exploration of novel sparse topologies. The initial sparse connectivity for SET is established using an Erdős-Rényi-Kernel (ERK) scheme, controlled by a parameter $\zeta$. SET was applied to all linear layers in our models, with $\varepsilon$, $\zeta$, and evolution frequency determined via ablations (Appendix D), while the fixed sparsity was kept at 95% for consistency with the GMP sparsity.

Table 5: Configuration details for gradual magnitude pruning, sparse evolutionary training, and the alternative plasticity-enhancing methods (ReDo, Reset, and Weight Decay) evaluated.

| Hyperparameter | Value |
|---|---|
| *Gradual Magnitude Pruning:* | |
| Desired Sparsity $\rho_F$ | 95% |
| Pruning Frequency $f_p$ | 500 timesteps |
| Pruning start interval $t_{\text{start}}$ | 0.05$\times$ number of timesteps |
| Pruning end interval $t_{\text{end}}$ | 0.80$\times$ number of timesteps |
| Sparsity $\rho_t$ at timestep $t$ | $\rho_F \left[ 1 - \left( 1 - \frac{t - t_{\text{start}}}{t_{\text{end}} - t_{\text{start}}} \right)^3 \right]$ |
| Prune Bias | False |
| Pruned Layers | [Conv2D, Linear] |
| *Sparse Evolutionary Training:* | |
| Sparsity | 95% |
| $\varepsilon$ density | 11 |
| Sparsity Distribution $\zeta$ | 0.3 |
| Evolution Frequency | 2000 timesteps |
| *Plasticity:* | |
| Plasticity Buffer Max Size | 100000 |
| ReDo (Sokar et al., 2023) Frequency $f_d$ | 5000 timesteps |
| Dormant Neuron Threshold $\tau$ | 0.001 |
| Dormant Activation Batch Size | 1024 |
| Fisher Trace Batch Size | 1024 |
| Effective Rank Batch Size | 1024 |
| Effective Rank Target | Shared Feature Extractor |
| Reset Frequency $f_r$ | 100000 |
| Number of Resets $m$ | 2 |
| Reset Target Layers | Output |
| Weight Decay Coefficient $\lambda$ | $1 \times 10^{-6}$ |

# D    Sparse Methods Hyperparameters

This appendix details the hyperparameter selection for GMP, with tuning experiments conducted on MTPPO with the MT5 benchmark. Figure 8 illustrates GMP ablations: the left subplot shows that initiating pruning early (e.g., at 5% of training) and concluding by 80% is favorable for the schedule window $[t_{\text{start}}, t_{\text{end}}]$; the center subplot indicates stable performance across moderate pruning frequencies $f_p$; and the right subplot explores various final target sparsities $\rho$. Figure 9 presents SET tuning: the left subplot suggests moderate evolution connection $\varepsilon$ (e.g., 11-15) perform well; the center explores sparsity distribution parameters $\zeta$ and the right shows less frequent evolution can be beneficial. These ablations informed the hyperparameter choices used in our main experimental evaluations. All tuning experiments were run with the original training configuration, outlined in Table 2 for 30 seeds and 200 epochs. For visual purposes, we omit the confidence intervals of the less successful runs to avoid cluttering and only include them for the best hyperparameter run.

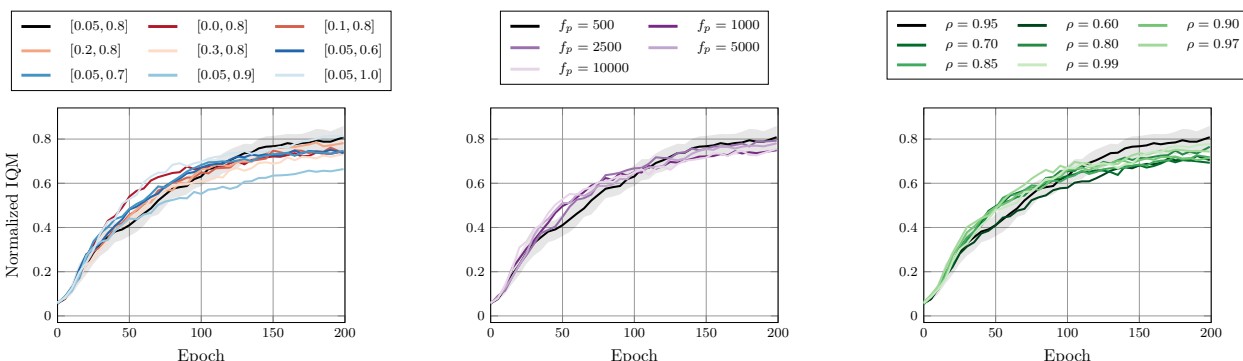

Figure 8: Hyperparameter ablation for Gradual Magnitude Pruning (GMP) on MTPPO with the MT5 benchmark. (Left) Varying pruning schedule window $[t_{\text{start}}, t_{\text{end}}]$. (Center) Different pruning frequencies $f_p$. (Right) Various final target sparsity levels $\rho$.

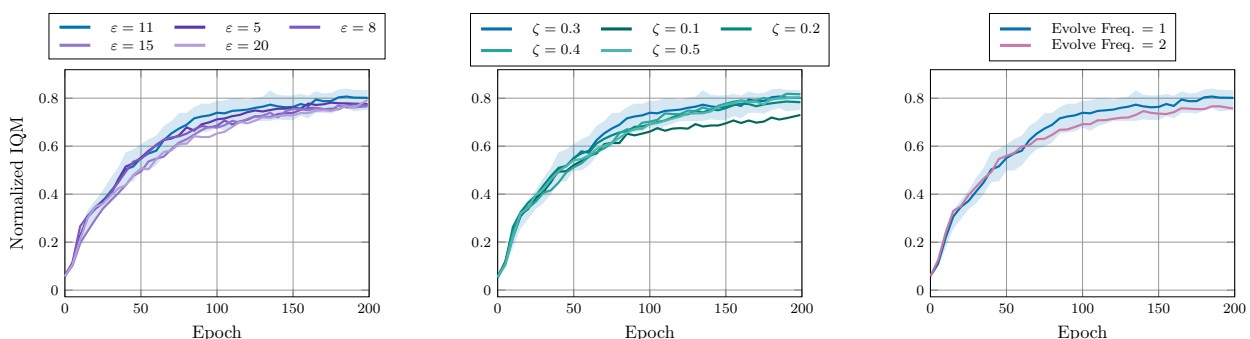

Figure 9: Hyperparameter ablation for Sparse Evolutionary Training (SET) on MTPPO with the MT5 benchmark. (Left) Varying evolution connection percentage $\varepsilon$. (Center) Different sparsity distribution parameters $\zeta$. (Right) Different evolution frequencies (episodes).

# E    Comparison Methods Hyperparameters

This appendix outlines hyperparameter considerations for the comparison methods evaluated in this work. All tuning experiments were primarily conducted on the MTPPO architecture using the MT5 benchmark to establish robust configurations. All tuning experiments were run with the original training configuration, outlined in Table 2 for 30 seeds and 200 epochs. For visual purposes, we omit the confidence intervals of the less successful runs to avoid cluttering and only include them for the best hyperparameter run.

For ReDo, tuning (Figure 10) highlighted the dormancy threshold $\tau$ as the most critical hyperparameter. Lower values, specifically $\tau = 0.001$ and $\tau = 0.0001$, demonstrated significantly better performance compared to higher thresholds or omitting the threshold entirely. The ReDo application frequency $f_d$ exhibited less sensitivity, with a moderate frequency (e.g., $f_d = 5000$ steps) performing well.

For the Reset mechanism, we investigated the impact of the reset frequency $f_r$ (how often resets occur) and the maximum number of times $m$ specific layers are reset throughout training, as illustrated in Figure 11. Our ablations indicated that a reset frequency of $f_r = 100k$ timesteps with a maximum of $m = 2$ resets per targeted layer (black line in both subplots) often provided a good balance between promoting plasticity and avoiding excessive training instability.

The Weight Decay (WD) coefficient $\lambda$ was selected from a standard range. As shown in Figure 12, very small coefficients (e.g., $\lambda = 10^{-6}$, black line) resulted overall in the best performance.

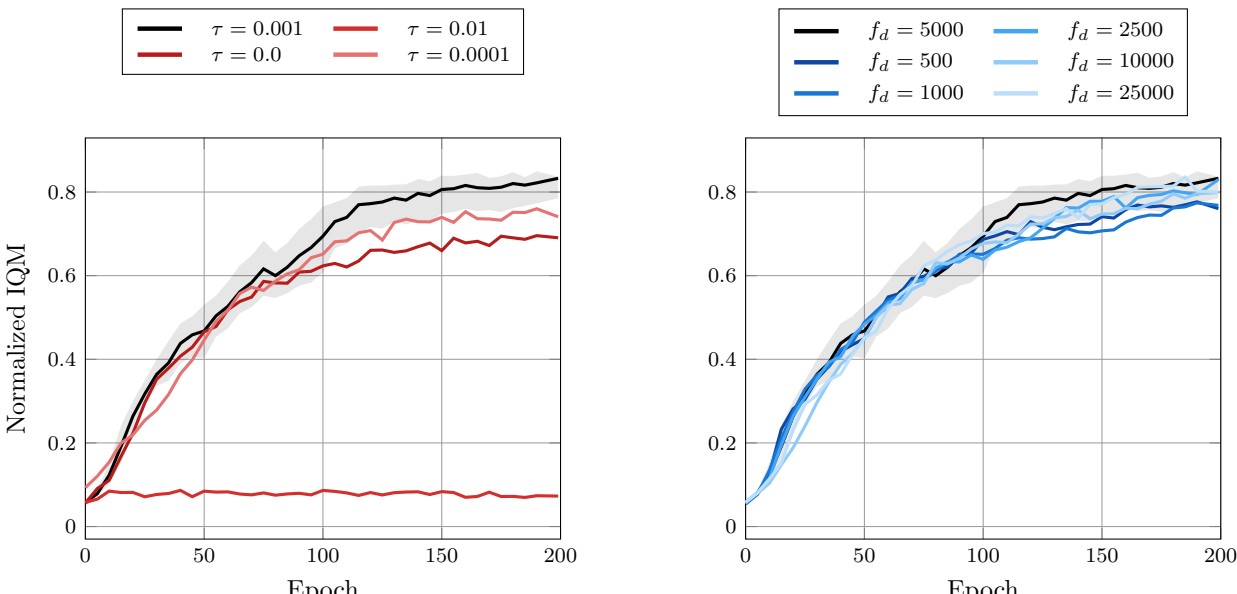

Figure 10: Hyperparameter ablation for ReDo on MTPPO with the MT5 benchmark. (Left) Varying dormancy threshold $\tau$. (Right) Different ReDo application frequencies $f_d$. The black line ($\tau = 0.001, f_d = 5000$) shows the best performance.

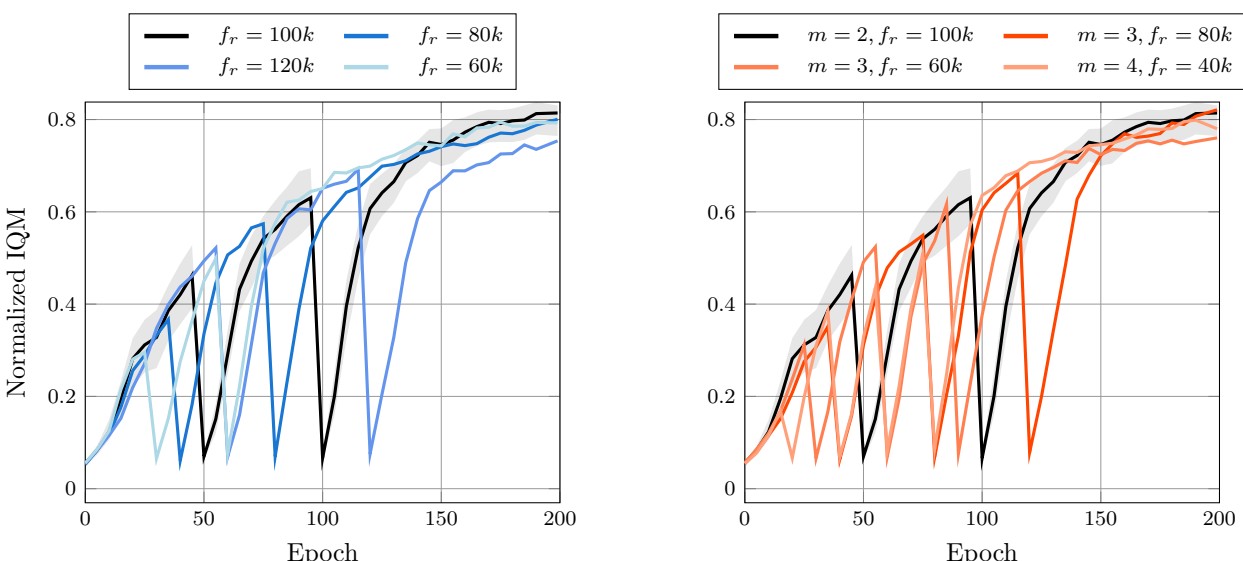

Figure 11: Hyperparameter ablations for the Reset mechanism on MTPPO with the MT5 benchmark. (Left) Varying reset frequency $f_r$ (with $m = 2$). (Right) Varying maximum number of resets $m$. The black line ($f_r = 100k, m = 2$) indicates the best performance.

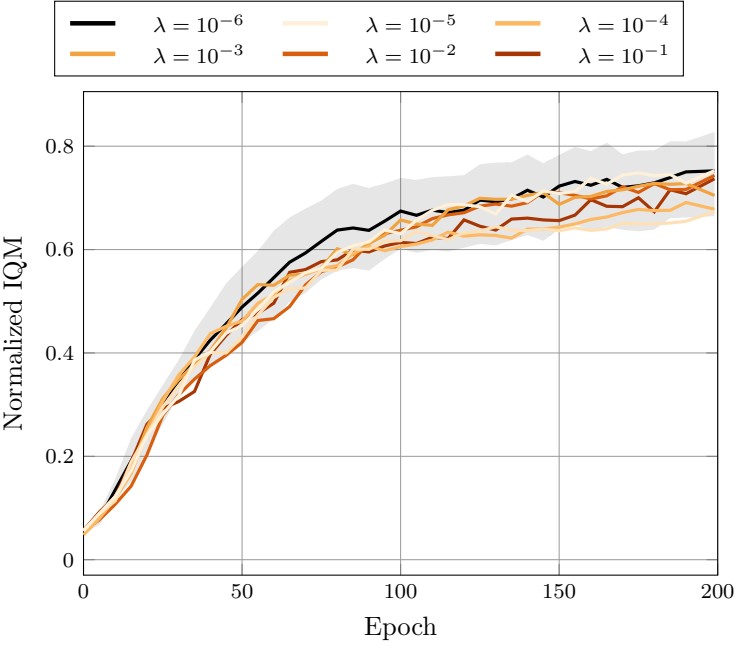

Figure 12: Hyperparameter ablation for Weight Decay (WD) on MTPPO with the MT5 benchmark, showing performance for different decay coefficients $\lambda$. The black line ($\lambda = 10^{-6}$) shows the best performance.

## F    Detailed Learning Curves

This appendix provides the complete learning curves for various agent configurations, complementing the aggregated final performance data presented in Table 1. These plots illustrate the training progression across all evaluated MTRL architectures (MTPPO, MoE, MOORE) on the MT3, MT5, and MT7 benchmarks. Additionally, Figure 13 presents the results obtained on the continuous control MetaWorld MT10 benchmark that complement the findings described in Section 4.1.

Figure 14 displays learning curves comparing our primary sparsification methods (Gradual Magnitude Pruning, GMP, at 95% sparsity, and Sparse Evolutionary Training, SET, at 95% sparsity) against a dense baseline and explicit plasticity-inducing interventions (ReDo and Reset).

Figure 15 similarly presents learning curves, in this case contrasting the same sparsification methods (GMP 95% and SET 95%) against a dense baseline and common implicit regularization techniques (Weight Decay – WD, and LayerNorm).

In all subplots within Figure 14 and Figure 15, the horizontal dashed line indicates the aggregated performance of single-task PPO (ST PPO) agents. Each ST PPO agent was trained separately on a single environment from the respective benchmark for the full 400,000 timesteps (the same total duration as the multi-task agents) across 30 runs. Consequently, this ST PPO performance should be viewed as a potentially **near-maximal** reference point from a single-task perspective, as multi-task agents faced the more challenging scenario of learning all tasks within a benchmark concurrently using the same total number of timesteps.

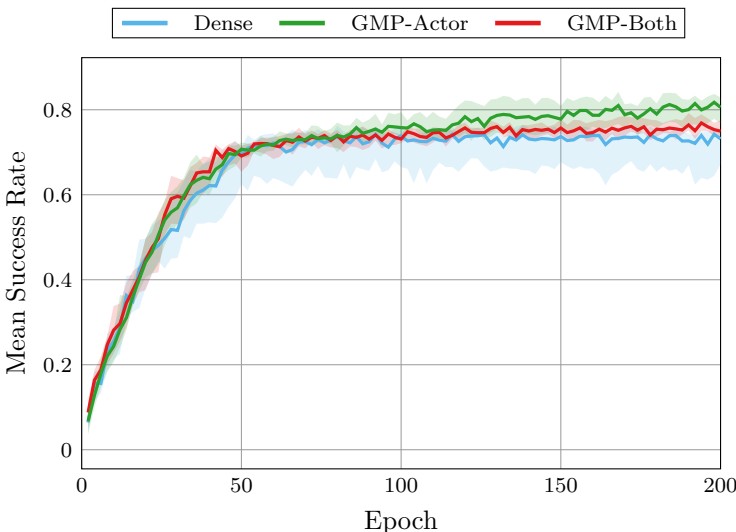

Figure 13: Final success rates on MetaWorld MT10 for different pruning configurations. Selective pruning of the actor network (GMP-actor) outperforms both the dense MTMH-SAC baseline and the globally pruned model (GMP-all), achieving a final success rate of 81%. This supports the hypothesis that actor-only pruning enhances performance by improving efficiency while preserving critical representational capacity in the critic.

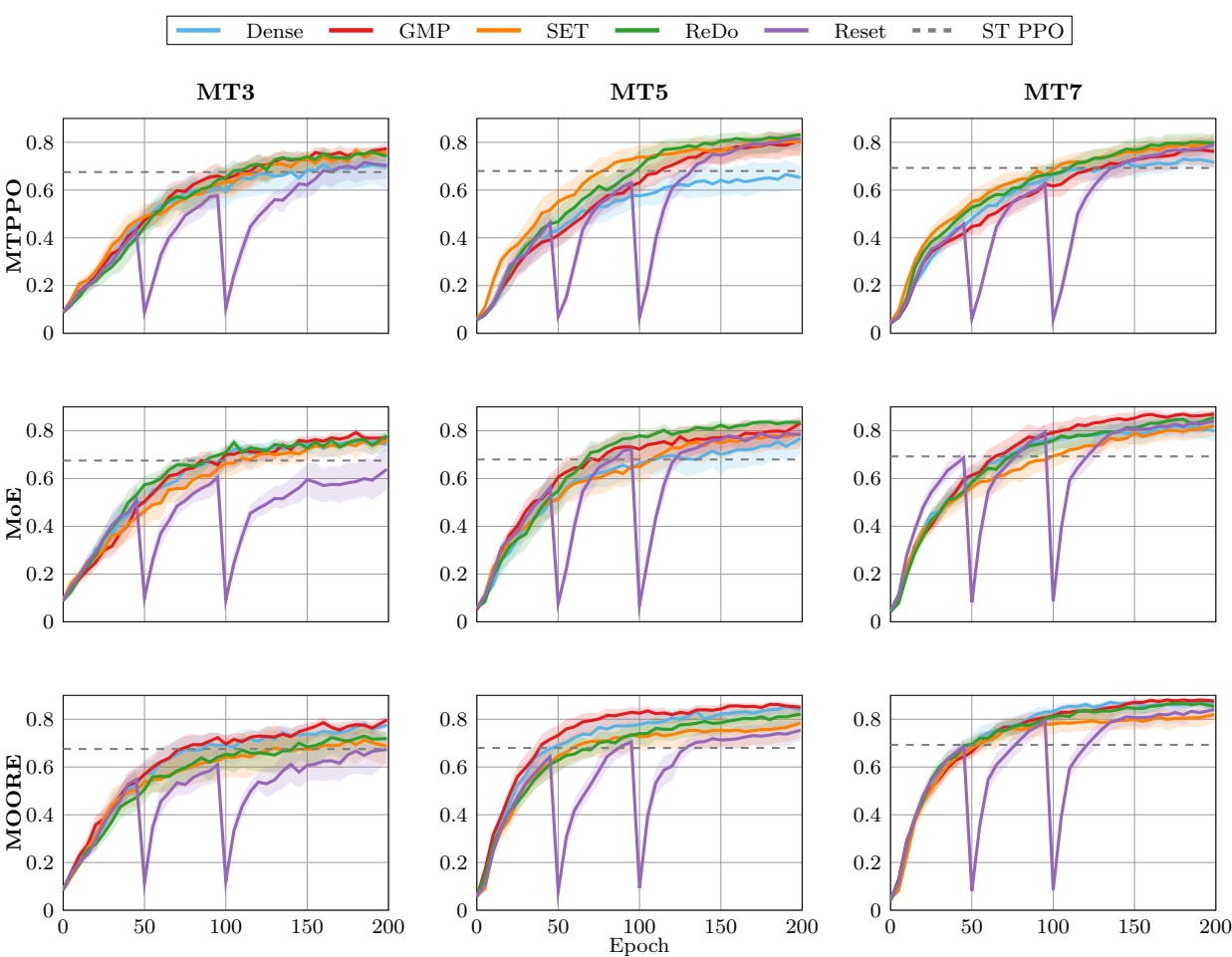

Figure 14: Learning curves (Normalized IQM) comparing sparsification methods (GMP 95%, SET 95%) with explicit plasticity-inducing interventions (ReDo, Reset) and a dense baseline. Results are shown for MTPPO, MoE, and MOORE architectures across MT3, MT5, and MT7 benchmarks. The dashed line represents single-task PPO performance.

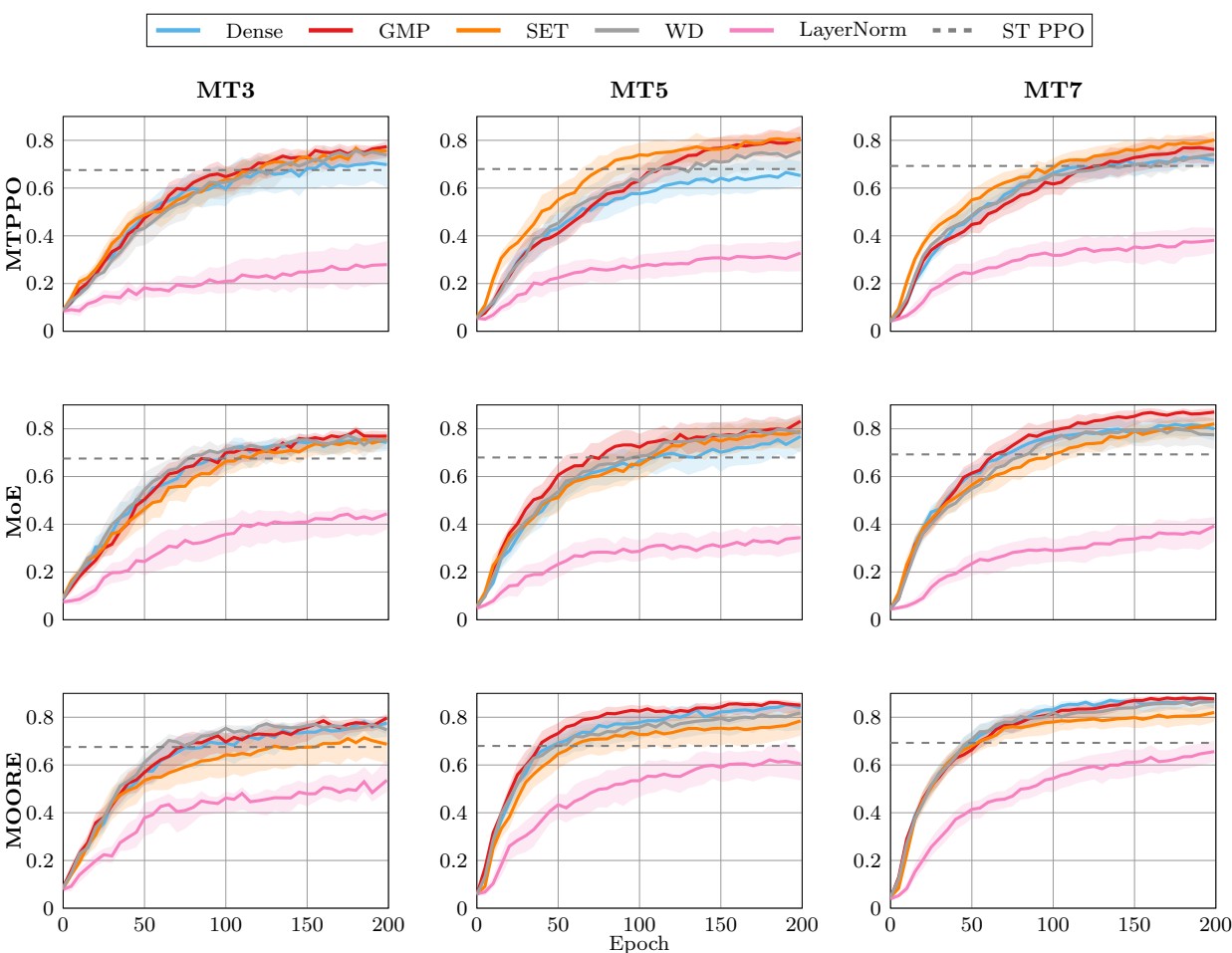

Figure 15: Learning curves (Normalized IQM) comparing sparsification methods (GMP 95%, SET 95%) with implicit regularization techniques (Weight Decay, LayerNorm) and a dense baseline. Results are shown for MTPPO, MoE, and MOORE architectures across MT3, MT5, and MT7 benchmarks. The dashed line represents single-task PPO performance.

# G   Detailed Plasticity Metrics

This appendix provides a comprehensive view of the plasticity metric evolutions across different benchmarks, complementing the primary analysis presented in Section 4.2 (which predominantly features results from the MT5 benchmark, also shown here for completeness as Figure 17 and Figure 20). The following figures illustrate the dynamics of Fisher Trace, Effective Rank, Actor Dormant percentage, and Critic Dormant percentage for all evaluated MTRL architectures (MTPPO, MoE, MOORE).

Figures 16, 17, and 18 compare the effects of sparsification methods (GMP and SET) against explicit plasticity-inducing interventions (ReDo and Reset) and a dense baseline, on the MT3, MT5, and MT7 benchmarks, respectively. While specific magnitudes vary, general trends such as SET's strong effect on reducing dormancy and GMP's characteristic Fisher Trace dynamics are often observable across benchmarks, though interactions with architecture (especially MOORE) can modulate these effects.

Similarly, Figures 19, 20, and 21 present a comparison of the same sparsification methods (GMP and SET) against implicit regularization techniques (Weight Decay – WD, and LayerNorm) and a dense baseline, for the MT3, MT5, and MT7 benchmarks, respectively.

Figure 22 presents the plasticity profiles on MetaWorld MT10, using the MTMH SAC architecture. The Critic MER and Critic Dormant neuron percentages are nearly identical across all three configurations, with dormancy already close to zero even for the dense baseline, leaving little room for improvement. The actor metrics, however, show clear distinctions. Selectively pruning the actor resulted in maintaining the lowest percentage of actor dormant neurons and an increased mean effective rank compared to both the dense baseline and the globally pruned agent. The Fisher Trace remained highly volatile for all methods and showed no discernible pattern.

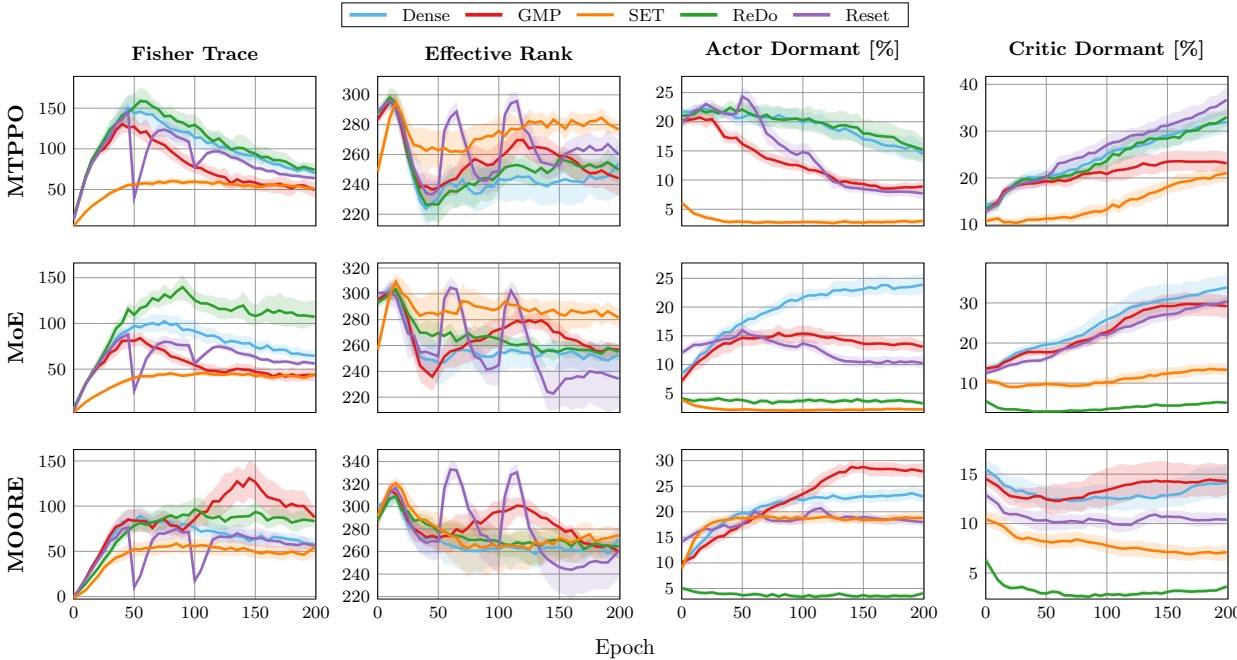

Figure 16: Comparative plasticity dynamics of sparse methods (GMP, SET) versus explicit plasticity-inducing interventions (ReDo, Reset) and a dense baseline, across MTPPO, MoE, and MOORE architectures on the **MT3** benchmark. Metrics include Fisher Trace, Effective Rank, and percentage of Actor and Critic Dormant Neurons.

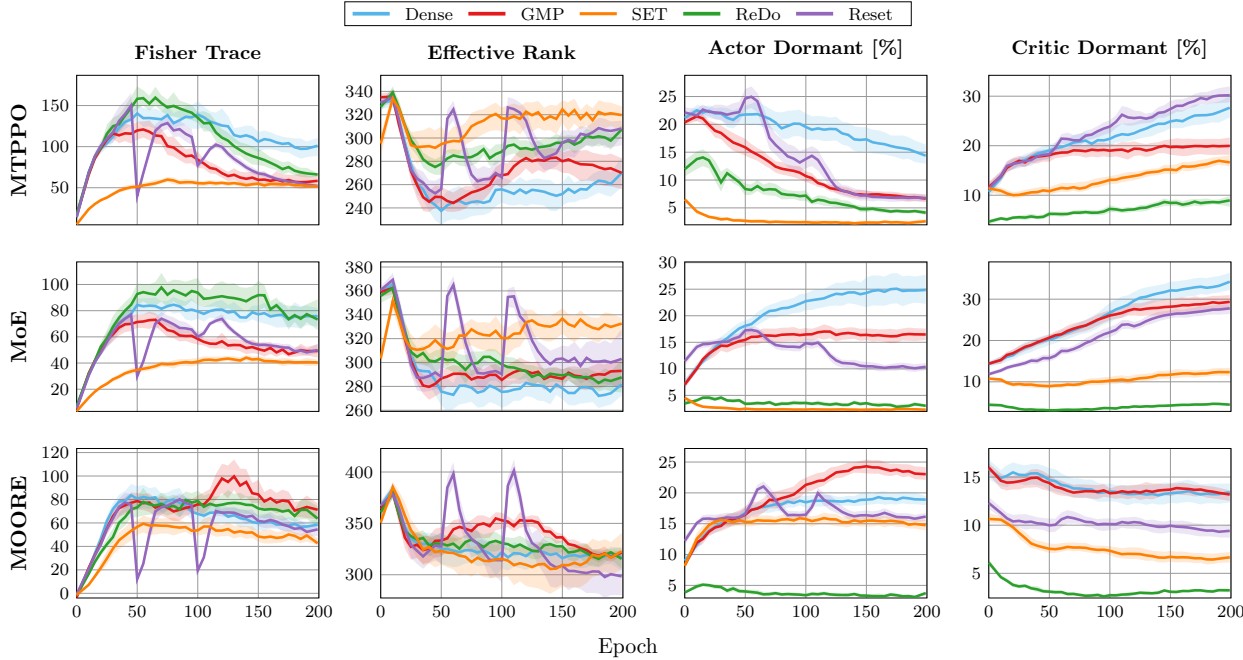

Figure 17: Comparative plasticity dynamics of sparse methods (GMP, SET) versus explicit plasticity-inducing interventions (ReDo, Reset) and a dense baseline, across MTPPO, MoE, and MOORE architectures on the **MT5** benchmark. Metrics include Fisher Trace, Effective Rank, and percentage of Actor and Critic Dormant Neurons.

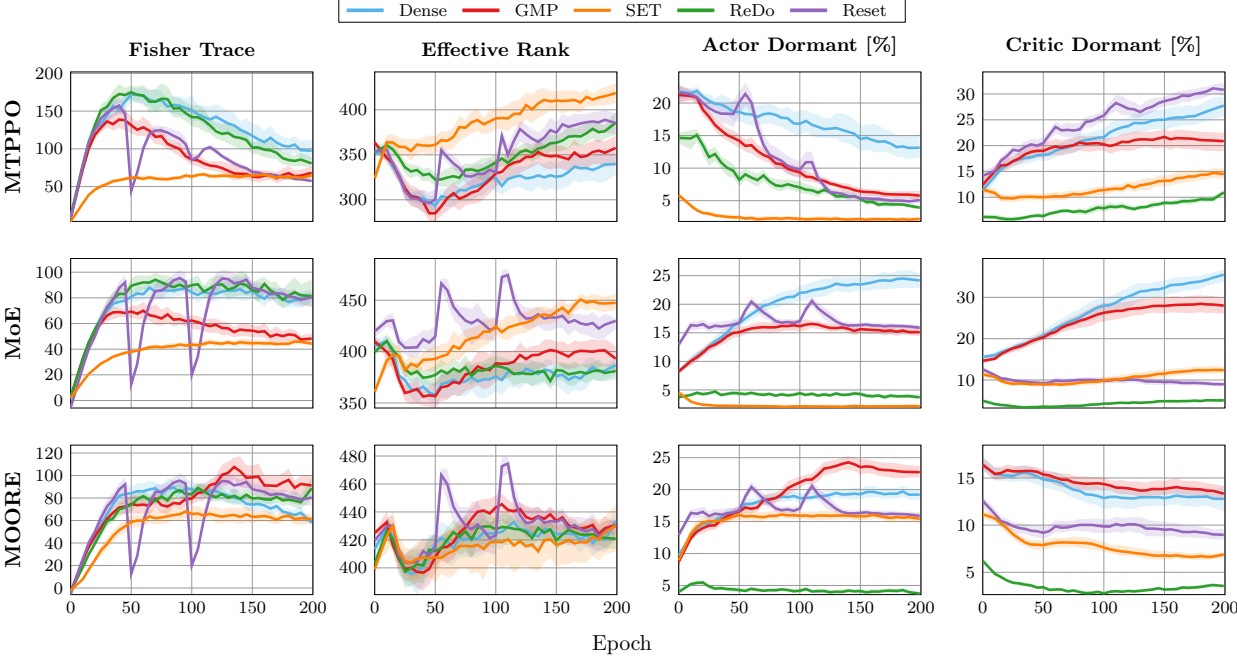

Figure 18: Comparative plasticity dynamics of sparse methods (GMP, SET) versus explicit plasticity-inducing interventions (ReDo, Reset) and a dense baseline, across MTPPO, MoE, and MOORE architectures on the **MT7** benchmark. Metrics include Fisher Trace, Effective Rank, and percentage of Actor and Critic Dormant Neurons.

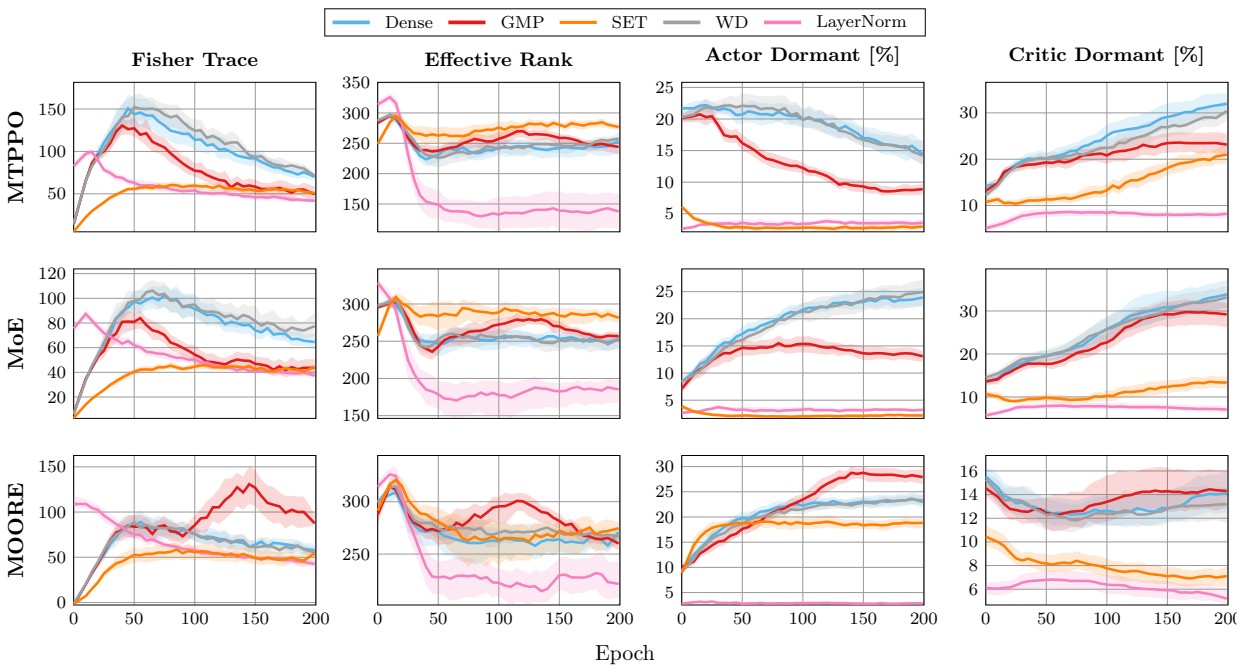

Figure 19: Comparative plasticity dynamics of sparse methods (GMP, SET) versus regularization techniques (Weight Decay, LayerNorm) and a dense baseline, across MTPPO, MoE, and MOORE architectures on the **MT3** benchmark. Metrics include Fisher Trace, Effective Rank, and percentage of Actor and Critic Dormant Neurons.

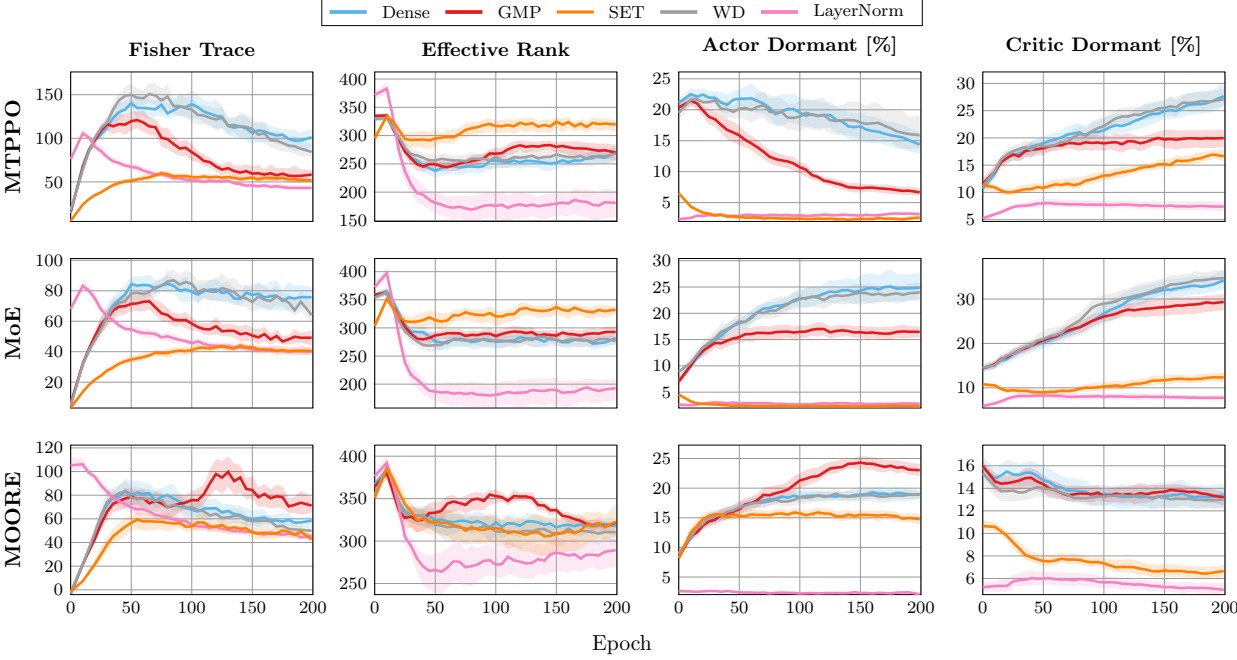

Figure 20: Comparative plasticity dynamics of sparse methods (GMP, SET) versus regularization techniques (Weight Decay, LayerNorm) and a dense baseline, across MTPPO, MoE, and MOORE architectures on the **MT5** benchmark. Metrics include Fisher Trace, Effective Rank, and percentage of Actor and Critic Dormant Neurons.

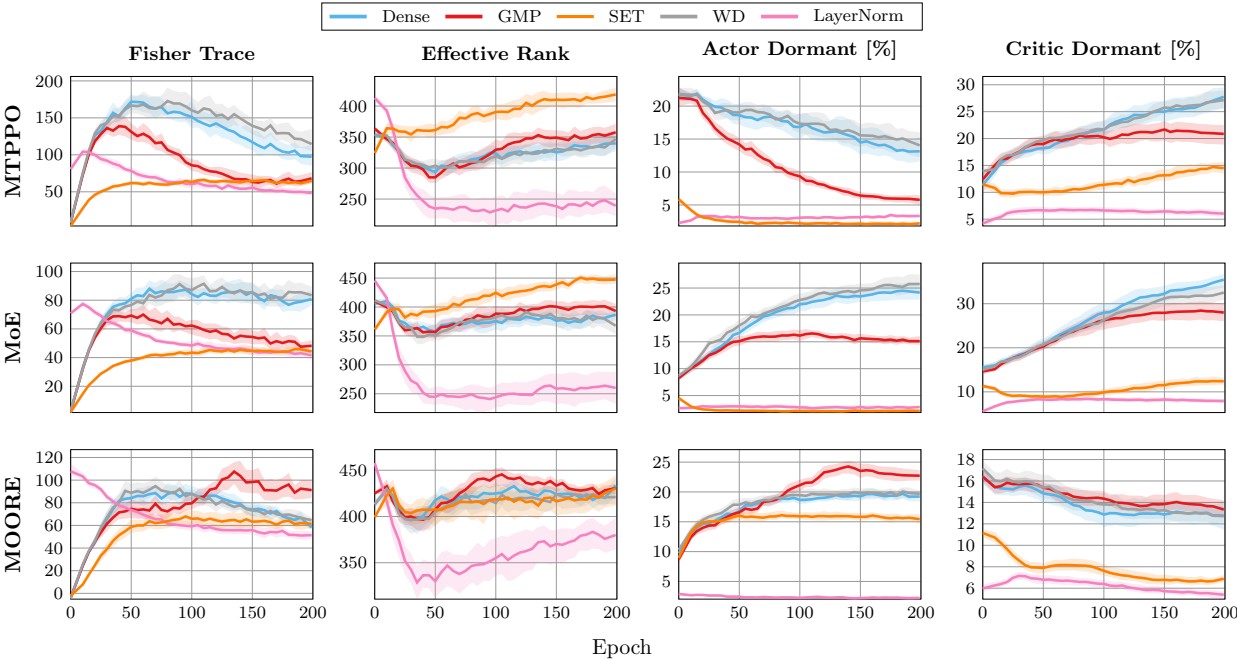

Figure 21: Comparative plasticity dynamics of sparse methods (GMP, SET) versus regularization techniques (Weight Decay, LayerNorm) and a dense baseline, across MTPPO, MoE, and MOORE architectures on the **MT7** benchmark. Metrics include Fisher Trace, Effective Rank, and percentage of Actor and Critic Dormant Neurons.

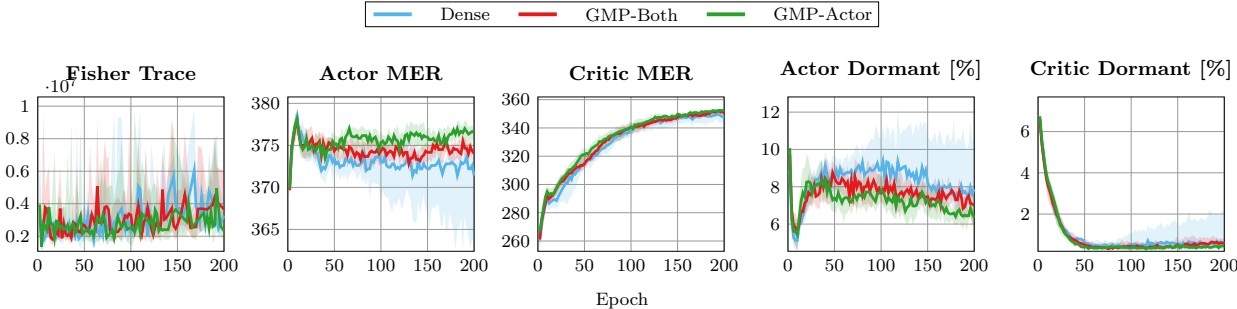

Figure 22: Plasticity dynamics on MetaWorld MT10. Selective pruning of the actor (GMP-Actor) leads to a decrease in dormant neurons and an increase in the mean effective rank compared to both the dense MTMH-SAC baseline and the globally pruned model (GMP-Both). In contrast, the critic metrics and Fisher Trace show minimal or no clear patterns.

# H   Gradual Magnitude Pruning Performance Across Sparsity Levels

This appendix presents learning curves for Gradual Magnitude Pruning (GMP) across various target sparsity levels (Dense, 80%, 95%, 99%) for all architectures and benchmarks (Figure 23). These results show that while 80% and 95% sparsity generally yield strong performance, often matching or exceeding dense baselines especially for MTPPO and MoE, the optimal sparsity level is architecture and benchmark-specific. Notably, for the MOORE architecture, 99% sparsity leads to performance degradation later in training (visible in Figure 23, bottom row). This performance drop correlates with a significant representational rank collapse, as illustrated by the sharp decline in effective rank for MOORE (Figure 24), indicating a loss of representational diversity under extreme pruning in this specific architecture.

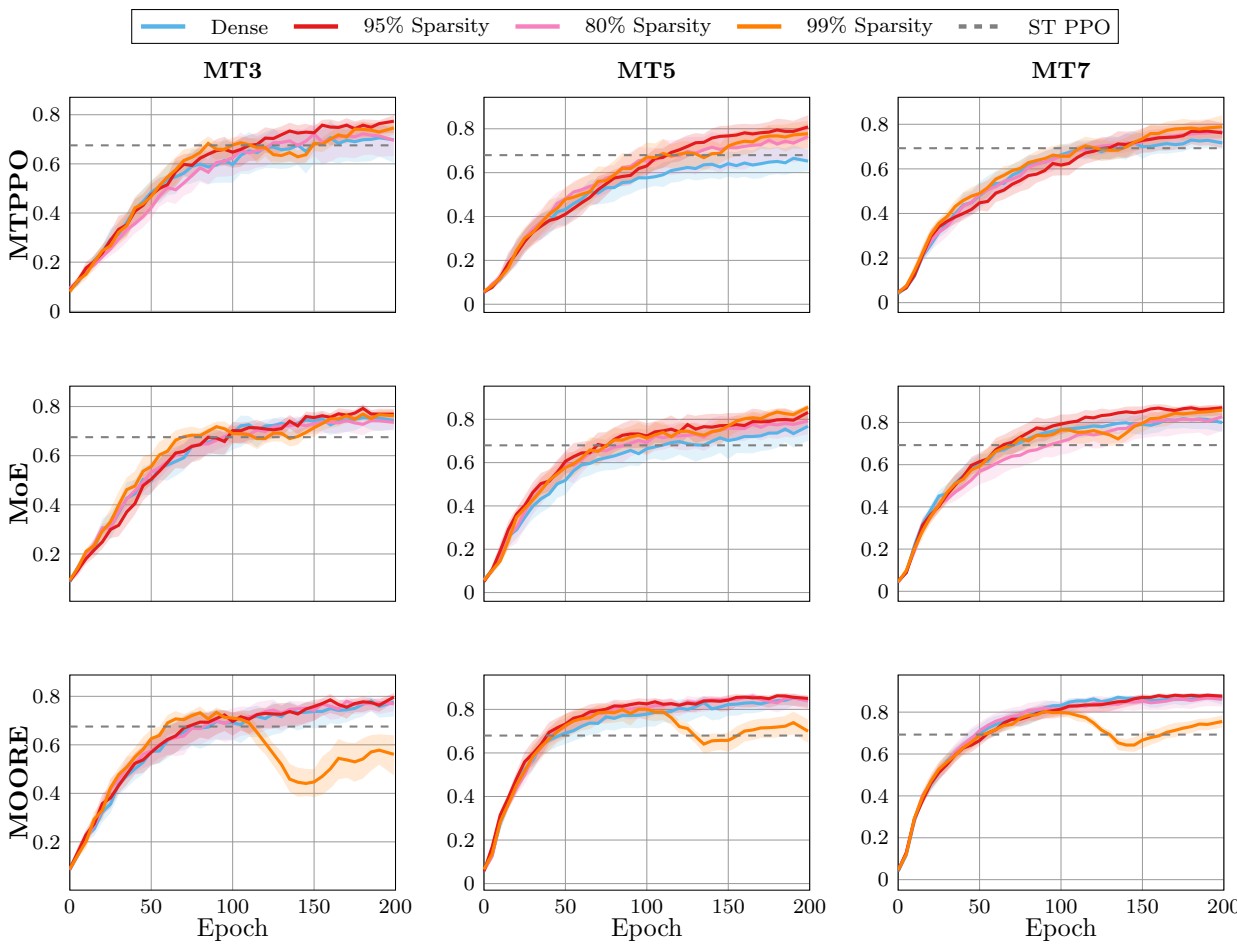

Figure 23: Normalized aggregate returns for agents under different GMP sparsity levels (Dense, 80%, 95%, 99%) across the MT3, MT5, and MT7 benchmarks for MTPPO (top row), MoE (middle row), and MOORE (bottom row) architectures. The dashed line represents single-task PPO performance.

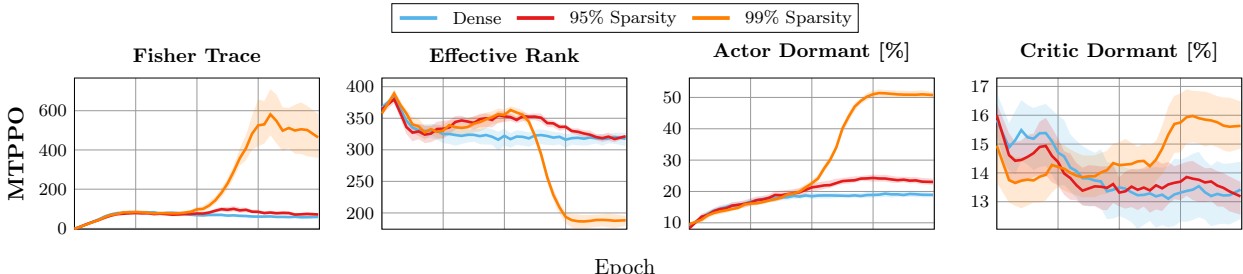

Figure 24: Plasticity metrics for MOORE on the MT5 benchmark under different GMP sparsity levels, illustrating the rank collapse at 99% sparsity, characterized by a sudden increase in the Fisher Trace and neuron dormancy, and a sharp drop in the effective rank.

