# OpenReview forum: "Sparsity-Driven Plasticity in Multi-Task Reinforcement Learning"
_TMLR — Accepted by TMLR_

### Review · Reviewer_GspR · 2025-06-03

**Summary Of Contributions:**

This paper demonstrates the impact of gradual magnitude pruning on multi-task reinforcement learning (MTRL) problems. This approach is applied to a range of architectures including networks with shared backbones, mixture of experts (MoE), and mixture of orthogonal experts (MOORE) models. Overall, it is shown that gradual plasticity can improve the achieved accuracy in MTRL problems. It can also be shown that, when measuring indicators of plasticity (such as neuron dormancy, effective rank, and the trace of the Fisher Matrix) gradual magnitude pruning can lead to more favourable regimes (lower dormancy, higher rank, lower Fisher trace). One major exception is the case of MOORE architectures which often showed reduced accuracies after training and unfavourable changes in Rank and the related Neuron Dormancy.

**Audience:**

Yes

**Broader Impact Concerns:**

- None -

**Claims And Evidence:**

No

**Requested Changes:**

I would request a number of significant changes to this paper. Note, the scale of changes is so large that I would advise that this paper be rejected and either completely overhauled or presented at a small workshop event. Nonetheless, I have my requested changes below. I hope that this review is clear and helpful in providing input and improving the work.

- **Textual**: The text of this work, especially the abstract, introduction, and discussion, should be more clear on the limitations and drawbacks of this work. In particular, the abstract and introduction make no mention of the failure of this method when applied in some cases (e.g. MOORE), and the discussion only lightly touches on the drawbacks of the specifically tested sparsification method.
- **Methods/Theory**: The method itself requires far more explanation and justification than is currently provided. At present the sparsification schedule is described without any mention of why this form is appropriate, and it is only rather indirect in stating that this specific sparsification schedule is exactly that of Zhu & Gupta (2017). Furthermore, the reasoning or theoretical basis for why one should expect that this sparsification schedule (or sparsification in general) should result in benefits for neuron dormancy, effective rank, or Fischer Trace are also missing.
- **Experiment Hyperparameters**: It is unclear how the hyperparameters for the various methods were selected. For example, the weight sparsification values of 80/95/99 seem to be rather arbitrary. More importantly, the hyperparameters of the baseline methods do not show whether alternatives were tested. For example the application of ReDo and Reset are carried out at predetermined timesteps but this could have been further optimised. Weight decay also appears to have a single value used (10^(-3)) but how this was selected? This should have ideally been tested across a range and perhaps even tuned for each architecture.
- **Performance Curves**: From examination of the performance curves (e.g. Figures 3 and 4) it appears that the training of the models was not carried out to convergence (they still appear to be training at the end of the 100 epochs). This makes it difficult to know which of the compared methods for MTRL ultimately converge to the highest accuracy. This means that the conclusions of outperformance of this method are also still somewhat open to change. Furthermore, Figure 3 appears to show that the sparsification levels chosen (80/95/99) might in fact be far too high in some cases. For example, it appears that for the MOORE models in the MT3 task that all of these sparsity levels are in fact ‘too sparse’. It is not clear how these sparsity values were chosen but it would be good to see the performance of networks over a much wider range of sparsity values.

**Strengths And Weaknesses:**

**Strengths**:
- This paper is quite concise and clear and therefore easy to read
- The set of comparisons being run across multiple model architectures is good practice
- Comparisons against alternative regularizers such as Weight Decay, ReDo, and Reset are appreciated
- The application of sparsity for MTRL is a nice, and seemingly promising, direction of research

**Weaknesses**:
- The utility and novelty of the proposed method is unclear. Not only does it appear relatively ineffective for MOORE models, even in other architectures the performance is often matched or exceeded by an alternative and simple regularizer. Furthermore, the gradual pruning method is precisely taken from an existing work.
- The proposed method is not theoretically justified or explained. Therefore the method seems to suddenly appear without much reasoning, justification, or explanation of its particular form and whether this sparsification schedule is important or not.
- The method's drawbacks are not discussed in sufficient detail. For example, this method requires prior setting of the number of epochs of training. Furthermore, the choice of a gradual sparsificaion vs a fixed sparsification is never discussed or explored. Wave-like dynamics appear in the plasticity dynamics (Figure 2) and it is unclear if these are due to training or the sparsification/pruning schedule.
- The ineffectiveness of this method when applied to MOORE is discussed but never explicitly investigated and explained. Therefore, any application to alternative architectures is somewhat left as an uncertain exporation for practitioners.
- The results, when examined more closely, seem to suggest that the hyperparameter search may have been far too restricted in order for these results to be fair for all methods and models. Training curves in particular seem to suggest that the sparsity levels chosen might be too large in some cases and perhaps too small in other. It is also unclear whether the hyperparameters of baselines were sufficiently tuned.

---

> ### Author Response · Authors · 2025-06-03
>
> We thank the reviewer for the thorough response and feedback and believe their suggestions will strengthen our contributions significantly. We acknowledge the assessment that our work requires substantial revision to meet publication standards. Nonetheless, we believe the identified issues are methodological rather than fundamental and can be addressed through further experimentation and better presentation.
>
> **Reviewer's concern**: _"The method seems to suddenly appear without much reasoning, justification, or explanation of its particular form and whether this sparsification schedule is important or not."_
>
> We acknowledge this gap in the presentation. We directly build upon findings from Graesser et al. (2022) and Obando-Ceron et al. (2024), who demonstrate the effectiveness of the proposed schedule in single-task Atari and MuJoCo. In particular, Graesser et al. showed that this specific pruning schedule consistently outperformed dense baselines, static pruning, and alternative sparsity methods (Sparse Evolutionary Training and Rigged Lottery approaches). Moreover, Obando-Ceron et al. directly propose the examination of gradual magnitude pruning within multi-task learning in their Future Work section.
> As a revision, we will substantially expand our introduction and methodology to further provide motivation for the choice of this specific pruning schedule while also addressing its drawbacks.
>
> **Reviewer's concern**:  _"Training of the models was not carried out to convergence."_
>
> Even though the training configuration was directly borrowed from Hendawy et al. (2024), all experiments will be extended (repeated) to 150 epochs until convergence according to the sample efficiency curves.
>
> **Reviewer's concern**: _"The hyperparameter search may have been far too restricted and sparsity values are rather arbitrary."_
> The 80% sparsity level is directly taken from Graesser et al. (2022), while the 95% and 99% values are adopted from Obando-Ceron et al. (2024). These choices were generally made according to existing literature of their effectiveness in the single-task setting. We will motivate this further in the introduction.
>
> Nonetheless, to draw robust conclusions, we will run the following ablation studies in order to address these limitations in across the three benchmarks and architectures:
> * Sparsity level analysis: Run the experiments with final sparsity levels of 60%, 80%, 85%, 95%, 97%, 99%.
> * Pruning schedule intervals: Out of the best performing sparsity levels, repeat the experiments with pruning timestep intervals of 250, 500, 1000, and 2500 (current 500).
> * Pruning window: Repeat the experiments with pruning starting at 0%, 5%, 10%, and 20% of the training timesteps, and ending at 80%, 90%, and 100% (current [5%, 80%]).
> * Hyperparameter sweeps for ReDo dormant thresholds (0, 0.025, 0.1, 0.3), ReDo resetting interval (every 500, 1000, 2500, 5000, 10000 timesteps), Reset intervals ([60k, 120k, 180k, 240k], [80k, 160k, 240k], [100k, 200k]), and weight decay parameters across orders of magnitude (10e-6 to 10e-1).
>
> **Reviewer's concern**:  _"The reasoning or theoretical basis for why one should expect that this sparsification schedule (or sparsification in general) should result in benefits for neuron dormancy, effective rank, or Fischer Trace are also missing"_
>
> We acknowledge that our current work maintains a strongly empirical focus rather than theoretical development. The observed Fisher trace dynamics can be explained by the peak-and-decline pattern, associated with flatter local minima resulting from sparse neural networks. While theoretical grounding is not the primary contribution of our work, we will try to enhance our experimental analysis section with more theoretically-focused connections to plasticity metrics.
>
> We believe the reviewer's concerns, while substantial, identify addressable limitations rather than fundamental flaws in our approach. We are committed to conducting this additional work to provide the necessary results to support our claims. We kindly ask the reviewer whether these proposed changes will allow the reviewer to consider our paper for a potential publication at TLMR.
>
> References:
>
> * Johan Obando-Ceron, Aaron Courville, and Pablo Samuel Castro. In value-based deep reinforcement learning, a pruned network is a good network, June 2024. URL http://arxiv.org/abs/2402.12479. arXiv:2402.12479
>
> * Laura Graesser, Utku Evci, Erich Elsen, and Pablo Samuel Castro. The State of Sparse Training in Deep Reinforcement Learning, June 2022. URL http://arxiv.org/abs/2206.10369. arXiv:2206.10369
>
> * Ahmed Hendawy, Jan Peters, and Carlo D’Eramo. Multi-Task Reinforcement Learning with Mixture of Orthogonal Experts, May 2024. URL http://arxiv.org/abs/2311.11385. arXiv:2311.11385

---

> > ### Comment · Reviewer_GspR · 2025-06-17
> >
> > Thank you for your response. Indeed your suggestions for changes would address almost all limitations. To make it very concrete, below are the changes I would expect and how your response meets or does not meet these:
> >
> > - **An overhaul of the text and methods sections to properly motivate the sparsification method and to give explanation of it's features and limitations**: Your response text is great but largely focussed upon features, so please also consider limitations.
> > - **Training of all models to convergence or with multiple repeats until statistical significance can be determined**: You have suggested 150 epochs but more epochs maybe necessary, please judge this based upon empirical testing. I cannot judge this until these results are available.
> > - **A more full and complete search over hyperparameters, especially for the baselines**: Your suggested hyperparameter search seems very reasonable and much more extensive than what you currently have.
> > - **The inclusion of a single alternative baseline in which sparsification is not gradual but static**: This is currently still missing from your response and I do believe it is a useful baseline to include.
> >
> > If I was to see all of these changes to a satisfactory degree, I'd be happy to adjust my scoring for this work and to recommend it's publication.

---

> ### Author Response · Authors · 2025-07-04
>
> We thank the reviewer for the thorough and detailed review, and for providing a clear and concrete roadmap for improvement. We have now revised the manuscript by addressing the concerns raised:
> - Overhaul of Text (Motivation & Limitations): We have overhauled the text as requested. The Introduction and Related Work (Sec 2.2) now provide clearer motivation for our methods. Crucially, we added a dedicated subsection, 6.1 "Limitations and Practical Considerations," to explicitly discuss the drawbacks and practical trade-offs of both GMP and SET. Moreover, our results presentation and discussion has changed significantly by including separate sections, comparing sparse methods against explicit and implicit plasticity interventions.
> - Training to Convergence: All experiments in the revised paper have been extended to 200 epochs to ensure convergence. All figures and tables reflect these longer, completed training runs.
> - Full Hyperparameter Search: We conducted the requested extensive hyperparameter search. Appendix D details ablations for GMP and SET over a range of sparsity levels and schedule parameters. Appendix E details the comprehensive tuning performed for all baselines (ReDo, Reset, and Weight Decay).
> - Inclusion of a Static Baseline: To address the request for a non-gradual baseline, we now include Sparse Evolutionary Training (SET) in all of our main experiments, which has a fixed sparsity throughout training. Moreover, we include a preliminary experiment in Figure 1, comparing GMP (gradual), SET (fixed sparsity with dynamic topology), against LTH-style rewinding, which represents a form of one-shot static pruning. SET similarly showed strong performance, correlating with improved plasticity measures. We sincerely thank the reviewer for suggesting this addition, as we believe it significantly strengthened the "Sparsity-Driven Plasticity" narrative of our paper, making it more complete.
>
> We believe these extensive changes directly address each of the four points outlined. We hope the revised manuscript now meets the conditions necessary for acceptance.

---

### Review · Reviewer_8sLK · 2025-06-06

**Summary Of Contributions:**

This paper investigates gradual magnitude pruning as a method to mitigate plasticity loss in multi-task reinforcement learning (MTRL). The work addresses how deep RL agents lose adaptability over time due to gradient interference, representational collapse, and neuronal saturation.

The authors make three key contributions:
- demonstrating that gradual magnitude pruning (up to 95% sparsity) effectively mitigates plasticity loss.
- showing that pruning-induced plasticity benefits directly translate to better multi-task performance.
- revealing that pruning benefits vary across MTRL architectures (MTPPO, MoE, MOORE).

The evaluation uses PPO agents on MiniGrid environments across three multi-task architectures, demonstrating that network sparsity can simultaneously achieve compression and plasticity enhancement for more efficient and adaptable RL agents.

**Audience:**

Yes

**Claims And Evidence:**

Yes

**Requested Changes:**

**Requested Changes:**

• **Methodological justifications**: The authors need to provide clearer empirical support for their design choices:
  - Compare gradual pruning versus training sparse from initialization to demonstrate why the gradual approach is necessary
  - Analyze key hyperparameters such as initial pruning epochs, final sparsity targets, and pruning intervals.
  - **Investigate combining gradual pruning with weight decay to determine if gradual pruning provides benefits beyond standard regularization.**
  - Recent work has shown that architectures with normalization layers (e.g., layer normalization, batch normalization) can inherently reduce plasticity loss issues [1,2,3]. It would be valuable to test architectures with these normalization layers to determine whether gradual pruning still provides additional benefits when plasticity problems are already reduced by normalization.

• **Enhanced experimental validation**: The authors should include at least one experiment on a more complex multi-task benchmark (e.g., Multi-task Meta-World or similar) to demonstrate that the findings generalize beyond the relatively simple MiniGrid environments. This is essential to validate the practical applicability of the approach for real-world scenarios.

• **Comprehensive related work section**: Expand the literature review to include more thorough coverage.
   - pruning / dynamic network related [4,5,6]
   - plasticity-related [7,8,9]

[1] CrossQ: Batch Normalization in Deep Reinforcement Learning for Greater Sample Efficiency and Simplicity, Bhatt et al, ICLR'24.

[2] Bigger, Regularized, Optimistic: scaling for compute and sample-efficient continuous control, Nauman et al, NeurIPS'24.

[3] SimBa: Simplicity Bias for Scaling Up Parameters in Deep Reinforcement Learning, Lee et al, ICLR'25.

[4] RLx2: Training a Sparse Deep Reinforcement Learning Model from Scratch, Tan et al, ICLR'23.

[5] The State of Sparse Training in Deep Reinforcement Learning, Graesser et al, ICML'22.

[6] Neuroplastic Expansion in Deep Reinforcement Learning, Liu et al, ICLR'25

[7] On warm-starting neural network training, Ash et al, NeurIPS'20. (original reset paper).

[8] Slow and Steady Wins the Race: Maintaining Plasticity with Hare and Tortoise Networks, Lee et al, ICML'24.

[9] Plastic Learning with Deep Fourier Features, Lewandowski et al, arXiv'24.

**Strengths And Weaknesses:**

**Strengths:**
- **Novel research direction:** The application of gradual magnitude pruning to multi-task reinforcement learning addresses a relatively unexplored intersection, providing new insights into how sparsity interventions can mitigate plasticity loss in MTRL settings where representational flexibility demands are particularly high.
- **Comprehensive architectural analysis:** The paper provides thorough empirical evaluation across three distinct MTRL architectures (MTPPO, MoE, MOORE), revealing architecture-dependent effects and providing clear insights into when and why sparsity interventions are most beneficial for different network designs.

**Weaknesses:**
- **Insufficient methodological analysis:**
    - The paper lacks justification for gradual pruning over alternative sparsity strategies. Critical questions remain unanswered: What happens if models are trained sparse from initialization? Are there benefits to different pruning schedules?
    - No exploration of combining pruning with weight decay, despite weight decay being a standard regularization technique. Given the computational overhead of gradual pruning, stronger evidence is needed that it provides benefits beyond simpler alternatives like weight decay.
    - Missing ablation studies on key hyperparameters and pruning design choices.
- **Limited experimental scope:**
    - MiniGrid environments are relatively simplistic and may not adequately validate the approach's effectiveness for complex real-world scenarios. The evaluation would benefit from at least one experiment on more challenging multi-task benchmarks (e.g., Multi-task Meta-World) to demonstrate broader applicability.
- **Incomplete related work:** The paper would benefit from more comprehensive discussion of relevant literature in network pruning for RL and recent advances in plasticity preservation methods.

---

> ### Author Response · Authors · 2025-06-08
>
> We thank the reviewer for their feedback and appreciate the overall positive assessment of our work's research direction. We are committed to including as much empirical evidence as possible based on the mentioned suggestions and will update the paper accordingly.
>
> **Reviewer's concern**: _"The paper lacks justification for gradual pruning over alternative sparsity strategies"_
>
> We acknowledge this methodological gap. We build upon findings from Graesser et al. (2022) and Obando-Ceron et al. (2024), who demonstrated the effectiveness of gradual magnitude pruning over static pruning and alternative sparsity methods in single-task settings. However, we recognize the need for more comprehensive justification in the multi-task context. As a revision, we will perform ablation studies comparing gradual pruning versus training sparse from initialization to demonstrate the potential benefit from the gradual approach.
>
> **Reviewer's concern**: _"No exploration of combining pruning with weight decay."_
>
> We will conduct additional experiments, combining gradual pruning with weight decay to determine if gradual pruning provides benefits beyond simpler alternatives like weight decay.
>
> **Reviewer's concern**: _"Missing ablation studies on key hyperparameters and pruning design choices."_
>
> We will conduct comprehensive ablation studies to address these limitations across all three benchmarks and architectures:
> * Sparsity level analysis: Run the experiments with different final sparsity levels.
> * Pruning schedule intervals: Out of the best performing sparsity levels, repeat the experiments with different pruning timestep intervals.
> * Pruning window: Repeat the experiments with pruning starting and ending at different percentages of the training timesteps.
> * Hyperparameter sweeps for ReDo dormant thresholds, ReDo resetting interval, Reset intervals, and weight decay parameters across orders of magnitude.
>
> **Reviewer's concern**: _"Incomplete related work section."_
>
> We will significantly expand the related works section, covering the pruning and dynamic sparsity network approaches, incorporating the suggested references on sparse training, and other regularization-based plasticity interventions.
>
> **Reviewer's concern**: _"Recent work has shown that architectures with normalization layers can inherently reduce plasticity loss issues."_
>
> We will further conduct experiments comparing the effectiveness of gradual pruning against layer normalization techniques, and the combination of both.
>
> We believe the reviewer's concerns identify important methodological limitations that can be addressed through the expanded experimental program outlined above. We are committed to conducting this additional work to provide comprehensive empirical support for our claims and to position our contribution more clearly within the broader landscape of plasticity intervention techniques in reinforcement learning.
>
> References:
>
> * Johan Obando-Ceron, Aaron Courville, and Pablo Samuel Castro. In value-based deep reinforcement learning, a pruned network is a good network, June 2024. URL http://arxiv.org/abs/2402.12479. arXiv:2402.12479
>
> * Laura Graesser, Utku Evci, Erich Elsen, and Pablo Samuel Castro. The State of Sparse Training in Deep Reinforcement Learning, June 2022. URL http://arxiv.org/abs/2206.10369. arXiv:2206.10369

---

> > ### Comment · Reviewer_8sLK · 2025-06-26
> >
> > Thank you for the detailed and thoughtful response. The proposed ablations and clarifications are valuable and will strengthen the paper.
> >
> > However, my main concern remains: the current evaluation is limited to MiniGrid, which limits the practical relevance and impact of the work. For the community to build on this, it’s important to demonstrate effectiveness on more realistic benchmarks.
> >
> > If the authors can show strong results on a benchmark like Meta-World MT-10 using a single architecture, I would be inclined to update my score positively.

---

> > > ### Author Response · Authors · 2025-07-02
> > >
> > > We sincerely thank the reviewer for their suggestion to conduct an additional experiment on the MetaWorld MT10 benchmark. We have now completed a full evaluation as recommended.
> > >
> > > Our experiments on MT10 were also motivated by the work of McLean et al. (2025), who showed that plasticity metrics were stable and dormant neuron counts were already low for the dense baselines on MT10. Notably, they demonstrate that scaling up the critic's capacity yields better performance than scaling up the actor's, confirming results from single-task RL. Building on this, we compare three configurations: (1) dense MTMH SAC as a baseline, (2) GMP applied to both the actor and the critic, and (3) GMP applied to the actor only.
> > >
> > > The results show that applying GMP to both components yields no significant performance gain over the dense baseline. However, applying GMP to the actor only results in increased performance in the task success rate, correlating directly with a reduction in the actor's dormant neurons. Those findings present a complementary perspective to McLean et al.'s work: while they showed the benefit of scaling the critic, our results demonstrate the benefit of pruning the actor and reinforce the principle of an asymmetric actor-critic relationship in reinforcement learning.
> > >
> > > We believe this new experiment is consistent with our findings on MiniGrid and adds a nuanced validation of the approach using a different benchmark and algorithm. We will update the manuscript with these findings shortly. We kindly ask the reviewer if the achieved results provide the necessary evidence to warrant the acceptance of the paper.
> > >
> > > McLean, Reginald, Evangelos Chatzaroulas, Jordan Terry, Isaac Woungang, Nariman Farsad, and Pablo Samuel Castro. "Multi-Task Reinforcement Learning Enables Parameter Scaling." arXiv preprint arXiv:2503.05126 (2025).

---

> > > > ### Comment · Reviewer_8sLK · 2025-07-02
> > > >
> > > > Thanks for the continued discussion.
> > > >
> > > > As an advocate for rigorous empirical science, my initial concern was that the paper's core contribution—the correlation between plasticity improvements from sparsification and enhanced multi-task performance—was solely demonstrated in MiniGrid. For a claim involving "improved performance," I felt broader empirical validation was crucial.
> > > >
> > > > The new results on Meta-World MT-10 are precisely what was needed. The finding that applying GMP to only the actor increases performance and reduces dormant neurons powerfully demonstrates the approach's effectiveness on a more realistic and challenging benchmark.
> > > >
> > > > Given these results from the MT-10 evaluation, I'm updating my recommendation.

---

> ### Author Response · Authors · 2025-07-04
>
> We thank the reviewer for positively updating their recommendation. We have now added Section 4.3, "Generalization to Continuous Control," which details the new experiment on the MetaWorld MT10 benchmark. As discussed, the finding that selectively pruning the actor network improves performance and plasticity (Fig 3) provides the requested validation on a more complex, continuous control benchmark. The full learning curves and plasticity metrics are presented in Appendix F and Appendix G and full training details are available in Table 3.

---

### Review · Reviewer_BVba · 2025-06-11

**Summary Of Contributions:**

The paper claims to make three main contributions, which are clearly stated in the abstract and the introduction.
1. In multi-task RL, gradual pruning mitigates some of the correlates of plasticity loss, such as dormant neurons.
2. Gradual pruning improves performance over dense networks and some plasticity-preserving methods like ReDo.
3. The effect of gradual pruning on plasticity and performance is architecture-dependent. And provide insights on when and why sparsity interventions help.

**Audience:**

Yes

**Claims And Evidence:**

No

**Requested Changes:**

Two major changes are needed to accept this paper:
1. Make results statistically significant. The authors already do 30 runs, but to show the difference between algorithms, more runs are needed, perhaps 100. The enviroments are simple and the networks are small, I think it would not require too much additional compute.
2. Tune baselines. More hyper-parameter setting for ReDo and resets need to be tested to claim that they perform worse than gradual pruning.

**Strengths And Weaknesses:**

The paper is well-written, it is easy to follow and understand. The key idea presented in the paper, gradual pruning, is easy to implement and does not require too much additional computation. The experiments are clearly designed and directly show if the claims made in the paper are true or not. The results support the first claim.

Despite these advantages, the paper has some fatal flaws.
1. **Performance results are not statistically significant.** The second claimed contribution of the paper is that gradual pruning improves performance over dense networks and some plasticity-preserving. This claim is based on the results presented in Table 1. However, Table 1 shows no statistically significant difference between the performance of dense or ReDo or Resets or gradual pruning. This means that claim 2 is not supported by the evidence provided.
2. **Baselines are not tuned.** A part of the second claim of the paper is that gradual pruning performs better than plasticity-preserving methods ReDo and resets. To show this, Redo and resets must be tuned for these environments, and a wide range of their hyper-parameters should be tested. Table 4 shows that only one value of their hyper-parameters is used.

In Addition, there are a few minor issues:
1. The second line of the title only contains "ing"; that is not aesthetically pleasing; consider moving the words "Reinforcement Learning" to the second line.
2. Chung et al. (2024) showed that maintaining orthogonality in weight matrices is a very effective technique for maintaining plasticity. I think the good performance of MOORE and the bad effect of ReDo and resets could be because MOORE encourages weights and representations to remain orthogonal. I suggest adding a discussion to contextualize the work of Chung et al. (2024).
3. The second half of the third contribution stated in the introduction ("revealing insights into when and why sparsity interventions can complement existing architectural configurations for multi-task learning") is not well supported. After reading the paper, I don't clearly see where the insights for "when and why sparsity interventions can complement existing architectural configurations for multi-task learning" are presented. Perhaps removing this part of the third contribution would be a good idea.

Chung et al. Parseval Regularization for Continual Reinforcement Learning. NeurIPS 2024.

---

> ### Author Response · Authors · 2025-06-12
>
> We sincerely thank the reviewer for their constructive feedback and kind words regarding the overall quality of our paper. We are particularly grateful for the appreciation of the simplicity of our proposed approach and for recognizing its potential relevance to the community.
>
> **1. Performance results are not statistically significant**
> We have taken this point seriously and have already acted on a similar concern raised by Reviewer GspR, who noted that *“training of the models was not carried out to convergence.”* In response, we have extended training from 100 to 200 epochs, which has already yielded more stable and statistically robust results. Additionally, we have increased the number of independent training runs to better capture variability. We are also in the process of incorporating more rigorous statistical tests to strengthen the empirical support for our findings.
>
> **2. Baselines are not tuned**
> The baseline results presented in the original submission were based on hyperparameter settings either recommended in prior work or derived from preliminary experimentation. However, we agree with the reviewer that a more exhaustive ablation study and hyperparameter tuning would improve the paper. This concern was also noted by reviewers 8sLK and GspR, the latter pointing out that *“the hyperparameter search may have been far too restricted and sparsity values are rather arbitrary.”* In line with our response to GspR, we are already expanding our experimental section to include more comprehensive ablations and a more careful tuning of the baseline configurations.
>
> Once again, we appreciate the reviewer’s overall positive stance on our work and the belief that it could be of interest to the research community. We are committed to addressing the experimental concerns in a timely manner and will incorporate all these improvements in the updated version of the paper.
>
> Finally, we thank the reviewer for pointing out minor issues. We will happily address them and will especially focus on the missing relevant literature. We will address this latter point alongside the related comments raised by Reviewer 8sLK.

---

> ### Author Response · Authors · 2025-07-04
>
> We sincerely thank the reviewer again for the clear and actionable feedback, which helped strengthen our paper significantly. We believe we have addressed the raised points through the following:
> - Statistical Significance & Training Convergence: To address this point, all experiments have been extended to 200 epochs to ensure convergence. Moreover, we reduced the level of our claims in the case when there is no statistical significance.
> - Baseline Tuning: We have conducted extensive hyperparameter sweeps for all baselines and methods, including ReDo and Reset. The detailed ablation studies and best-performing configurations for these methods can now be found in Appendix D and Appendix E.
> - Minor Issues: The title has been reformatted as suggested and we have incorporated the suggestion to discuss Chung et al. (2024).
>
> We thank the reviewer once again, and kindly ask if the revised manuscript addresses all the reviewer's concerns to make the paper acceptable at TLMR.

---

> > ### Comment · Reviewer_BVba · 2025-07-09
> >
> > Thank you for making the changes. I appreciate the effort you have put in doing additional experiments and aligning your claims with the results of the experiments.

---

### Author Response · Authors · 2025-07-04

We have uploaded the final version of our revised manuscript. This is our second revision, incorporating extensive changes based on the valuable feedback and discussion with the reviewers.

In addition to the manuscript updates, we have now posted detailed, final responses to each individual reviewer, demonstrating how their specific concerns have been addressed in this latest version.

We believe these comprehensive revisions, which include expanding our experimental scope to several new methods and benchmarks, performing extensive hyperparameter tuning for all methods and baselines, and substantial textual clarifications, fully address all points raised.

We are grateful for the reviewers' time and guidance, which has significantly strengthened this work. We hope the manuscript is now acceptable for publication at TMLR.

---

### Decision · Action_Editor_guR6 · 2025-07-15

**Recommendation:** Accept as is

**Audience:**

Yes

**Audience Explanation:**

Loss of plasticity is currently a major topic of interest in the machine learning community, clearly making this paper interesting to a considerable number of people in the community.

**Claims And Evidence:**

Yes

**Claims Explanation:**

The manuscript investigates loss of plasticity in multi-task reinforcement learning. When first submitted, its claimed contributions centered on demonstrating that pruning mitigates some trends correlated with plasticity loss and leads to performance improvement. This initial submission was followed by an analysis of different architectures and a discussion around when pruning is expected to help.

The first version of the manuscript had many significant issues raised by the reviewers, including concerns about presentation, a lack of statistical significance in the claimed results, untuned baselines, and missing important related work. The extent of the required changes was quite large, and it would have been comprehensible if the reviewers had decided the paper wasn't ready for publication. Nevertheless, the reviewers diligently evaluated the new submission and unanimously recommended its acceptance.

---

> ### Author Response · Authors · 2025-07-26
>
> We have uploaded the final camera-ready version of our manuscript.
>
> We would like to express our final thanks to the Action Editor and all the reviewers for their feedback and guidance. We are grateful for the constructive nature of the TMLR review process, which significantly improved the quality and scope of our work.